# Iterative taxonomic study of *Pareiorhaphis hystrix* (Siluriformes, Loricariidae) suggests a single, yet phenotypically variable, species in south Brazil

**Patrícia C. Fagundes, Edson H. L. Pereira, Roberto E. Reis**[ID]*

Pontifícia Universidade Católica do Rio Grande do Sul, PUCRS, Laboratory of Vertebrate Systematics, Porto Alegre, State of Rio Grande do Sul, Brazil

* reis@pucrs.br

**Data Availability Statement:** All relevant data are within the manuscript and its Supporting Information files, and in GenBank (accession numbers provided in S1 Table).

## Abstract

*Pareiorhaphis hystrix* is a widely distributed species, occurring in the upper and middle Uruguay River and in the Taquari River basin, Patos Lagoon system, southern Brazil. Morphological variation has been detected throughout the distribution of *P. hystrix*, and this work seeks to test the conspecific nature of populations in several occurrence areas. Specimens from six areas in the Uruguay River basin and three in the Taquari River basin were compared. Variance analysis (ANOVA) was performed for the meristic data, and Principal Component Analysis (PCA) and Linear Discriminant Analysis (LDA) were conducted for morphometric data. Molecular analyses used *col*, *cytb*, 12S and 16S mitochondrial genes, examining nucleotide diversity, haplotype diversity, genetic distance, and delimitation of possible multiple species through the Generalized Mixed Yule Coalescent (GMYC) method. Phylogenetic relationships of studied populations were also investigated through Bayesian inference. While PCA indicated a tendency of overlap between areas, ANOVA and LDA detected a subtle differentiation between populations from the two hydrographic basins. Yet, both latter analyses recovered the population from Pelotas River, a tributary to Uruguay River, as more similar to populations from Taquari River, which is congruent to morphological observations of anterior abdominal plates. The molecular data indicated a nucleotide diversity lower than the haplotypic diversity, suggestive of recent expansion. The concatenated haplotype network points to slight differentiation between areas, with each locality presenting unique and non-shared haplotypes, although with few mutational steps in general. The species delimitation by coalescence analysis suggested the presence of a variable number of OTUs depending on the inclusion or exclusion of an outgroup. In general, the morphological data suggest a subtle variation by river basin, while the genetic data indicates a weak population structuration by hydrographic areas, especially the Chapecó and Passo Fundo rivers. However, there is still not enough differentiation between the specimens to suggest multiple species. The iterative analyses indicate that *Pareiorhaphis hystrix* is composed of a single, although variable, species.

**Funding:** First author received a master degree scolarship from CAPES (Brazilian Ministry of Education). RER received a research grant for paying molecular work, from Conselho Nacional de Desenvolvimento Científico e Tecnológico - CNPq (process #400166/2016-0). The funders had no role in study design, data collection and analysis, decision to publish, or preparation of the manuscript.

**Competing interests:** The authors have declared that no competing interests exist.

## Introduction

Among catfish families, Loricariidae is the most diverse with above 1,000 species currently valid [1], a number that continues to raise steadily. Fishes in this family, popularly known as Cascudos in Brazil, are widely distributed in neotropical freshwaters from the Pacific drainages of southern Costa Rica to northeastern Argentina, with six subfamilies currently recognized [1, 2].

*Pareiorhaphis* Miranda Ribeiro, 1918 was removed from the synonymy of *Hemipsilichthys* Eigenmann, 1889 by Pereira [3] to reflect the phylogenetic relationships of the Delturinae and Neoplecostominae [4]. *Pareiorhaphis* currently has 26 valid species [5] and is endemic to Brazil, occurring in main coastal river drainages of south, southeast and northeast Brazil, in addition to eastern versants tributaries to the Paraná and São Francisco rivers [5, 6]. Species of *Pareiorhaphis* inhabit streams of strong water current and rocky bottom, usually being abundant where they occur, with greater diversity in the Doce River and coastal rivers of Santa Catarina State [5, 7].

The genus was recently demonstrated to be monophyletic by Pereira & Reis [8], and its species show remarkable morphological variation in several aspects like body size (maximum size 34 mm SL in *P. nudula* (Reis & Pereira, 1999) to 116 mm SL in *P. azygolechis* (Pereira & Reis, 2002)), color pattern, secondary sexual dimorphism, nature of abdominal cover, head, snout, and lip shapes, and morphometric and meristic features [3, 8]. Even the hypertrophied cheek odontodes of adult males, which represent a synapomorphy for the genus [8] are highly variable in size, thickness, density, direction, and position on the sides of the head.

*Pareiorhaphis hystrix* (Pereira & Reis, 2002) (Figs 1 and 2) was described almost two decades ago from the upper Pelotas River, Uruguay River basin, south Brazil, and later had its distribution expanded, being currently known from the upper and middle Uruguay River and the Taquari River basin, a tributary to the Patos Lagoon system. Its wide and disjoint distribution across two river basins diverges of most *Pareiorhaphis* species, which are usually restricted to a single drainage basin or to small basins historically connected in recent geological history.

Over the years, specimens of *Pareiorhapis hystrix* were collected in different environments in southern Brazil and subtle phenotypic variation among individuals has been noticed regarding color pattern, size and position of hypertrophied odontodes, and size of the cheek fleshy lobe. The geographic distribution of such variation has not been investigated, and it is thus not clear whether these represent intraspecific variation or could indicate the presence of multiple species. The combination of phenotypical data with molecular markers in an iterative approach, as opposed to the traditionally used morphology or molecules alone, has been efficient in fish population studies, and their use can contribute to the identification of cryptic species [9, 10].

The objective of this study is to perform a comparison between populations of *Pareiorhaphis hystrix* in the Uruguay and Taquari river basins, southern Brazil, to test the existence of undetected cryptic species. The hypothesis tested is the co-specificity of *P. hystrix* in the different areas of occurrence, with the null hypothesis being that *P. hystrix* constitutes a single, widespread species, while the alternative hypothesis predicts that *P. hystrix* aggregates different cryptic and closely related species lineages.

## Material and methods

The concept of iterative taxonomy is used in the sense of Yeates et al. [11], where species boundaries are treated as hypotheses to be tested by the comparison of multiple lines of evidence analyzed iteratively, in opposition to integrative taxonomy, where different sources of evidence should be integrated and analyzed together.

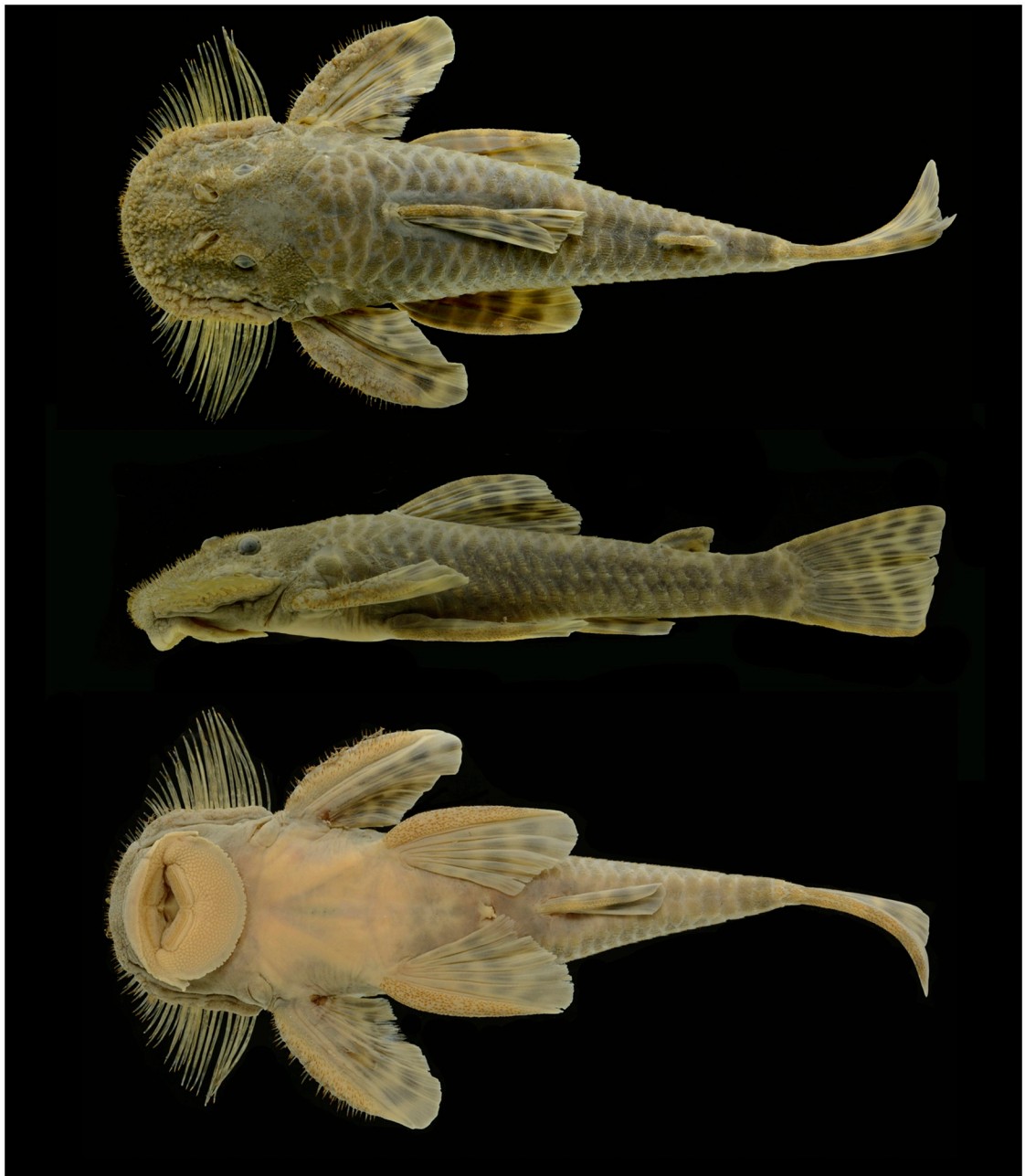

**Fig 1. *Pareiorhaphis hystrix*, MCP 48327, male, 99.2 mm SL in dorsal, lateral and ventral views.** Middle Antas River, Taquari River basin, São Francisco de Paula, Rio Grande do Sul State.

## Sampling and morphological analyses

Populations of *Pareiorhaphis hystrix* were sampled from six tributaries/sections of the Uruguay River (Chapecó, Pelotas, Ijuí, Passo Fundo, Middle Uruguay, and Canoas) and three of the Taquari River (Upper Antas, Middle Antas, and Prata) (Fig 3; Table 1). These tributaries correspond to study areas and fish samples are hereafter named according to the area, without implying they represent natural biological populations. All samples were collected under the collecting permit #10287 issued to RER by the Instituto Chico Mendes de Conservação da

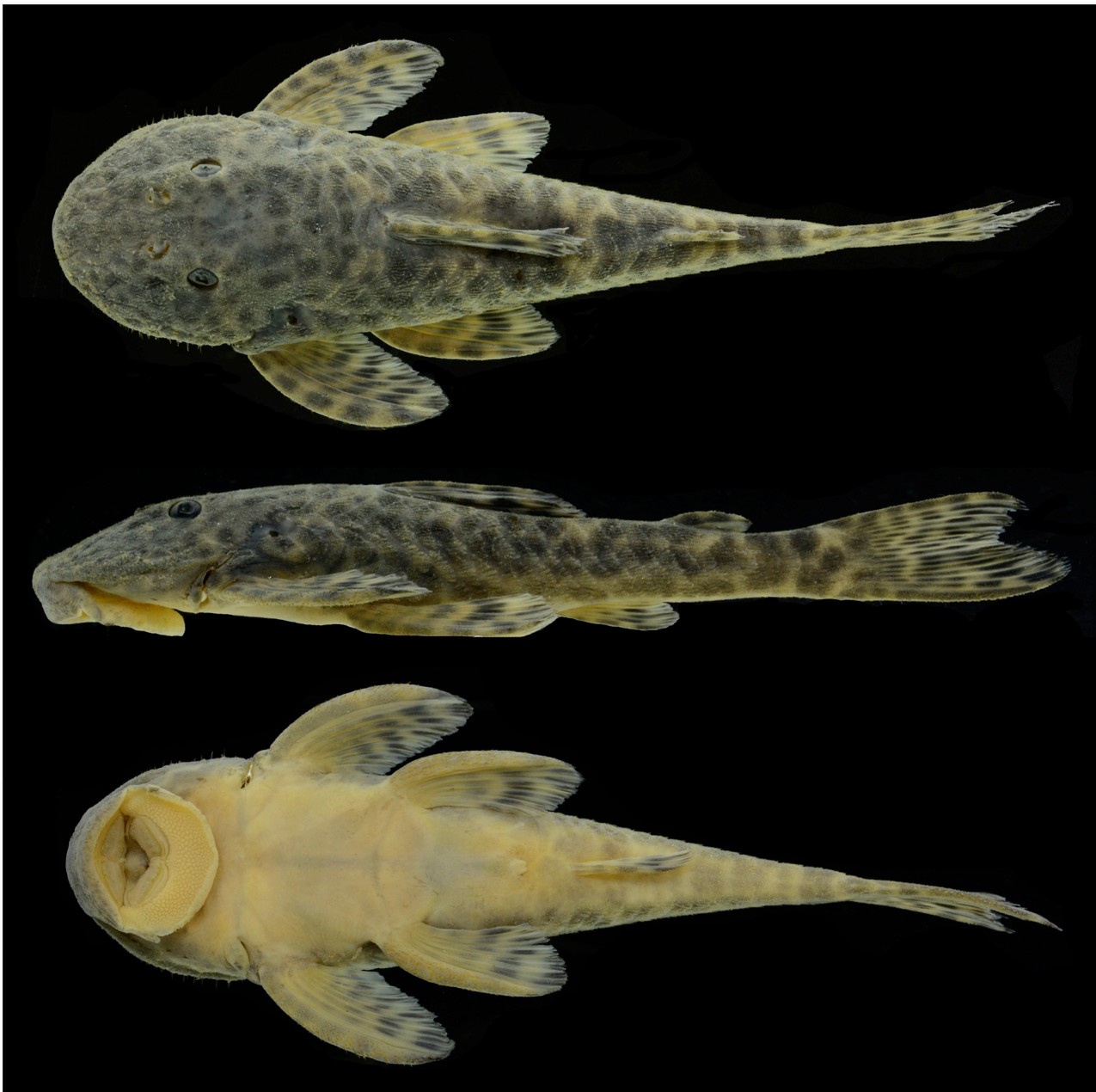

**Fig 2. *Pareiorhaphis hystrix*, MCP 53241, female, 85.0 mm SL.** Ponte Alta River, tributary to Canoas River, Uruguay River basin, Ponte Alta, Santa Catarina State.

Biodiversidade (ICMBio) of the Ministry of Environment. Fishes were euthanized through an over-exposition to a solution of clove oil, according to the protocol of Lucena et al. [12].

Morphological observations were made searching for variations on cheek fleshy lobe, hypertrophied male odontodes on cheek and pectoral-fin spine, color pattern, and abdominal plates with the specimens submerged in alcohol. Further morphological analyses were performed on adult specimens (above 60 mm) preserved in 70% ethanol. Counts and measurements were obtained according to Pereira et al. [7] from 211 specimens (S1 Table) representing each of the nine populations/areas, with body measurements being expressed as

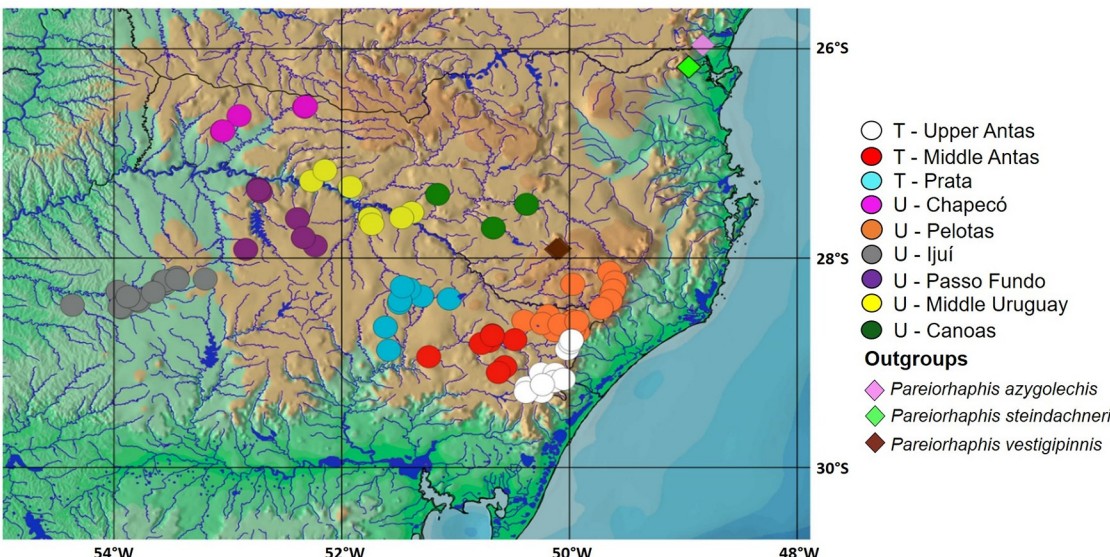

**Fig 3. Distribution of sampling localities of *Pareiorhaphis hystrix* and outgroups, including both alcohol specimens and tissue samples.** Each symbol may represent more than one lot or locality. T = Taquari River basin; U = Uruguay River basin.

percent of the standard length, while those of the cephalic region expressed as percent of the head length. The comparisons between the different areas were performed through statistical analyses of the morphometric and meristic values, presenting the minimum, maximum, mean, and standard deviation by area. The distribution map of Fig 3 was prepared using Quantum-GIS (v3.8), with shape and raster files available at IBGE (Instituto Brasileiro de Geografia e Estatística: http://mapas.ibge.gov.br/bases-e-referenciais), and ANA (Agência Nacional de Águas: http://www.snirh.gov.br/hidroweb), following the map preparation tutorial of Calegari et al. [13].

Power regressions ($y = a.SL^b$), where SL stands for standard length, were applied to morphometric data to convert measurements into residues and eliminate bias of individual size and allometric growth. Conversion was performed by nonlinear regression in SPSS v22.0 software using the Levenberg-Marquardt algorithm [14], seeking to eliminate the log-transformation bias. The same residues were compared morphometrically by multivariate analysis, Principal Component Analysis (PCA), which finds the main axes of variation of a dataset in any dimension, not discriminating whether these data belong to the same class) and Linear

**Table 1. Summary of area, river basin, number of lots, number of specimens, and number of tissue samples examined of *Pareiorhaphis hystrix*.**

| Area name | River basin | Lots | Specimens | Tissue samples |
|---|---|---|---|---|
| Upper Antas | Taquari | 21 | 34 | 8 |
| Middle Antas | Taquari | 15 | 25 | 5 |
| Prata | Taquari | 12 | 25 | 5 |
| Chapecó | Uruguay | 6 | 14 | 5 |
| Pelotas | Uruguay | 21 | 25 | 5 |
| Ijuí | Uruguay | 18 | 26 | 4 |
| Passo Fundo | Uruguay | 10 | 16 | 5 |
| Middle Uruguay | Uruguay | 13 | 22 | 5 |
| Canoas | Uruguay | 3 | 24 | 5 |

Discriminant analysis (LDA), which considers the existence of classes in the data, highlighting a linear separation in case it exists), using PAST v3.12 software [15]. Analysis of variance (ANOVA), box plots, and Tukey's tests were used to compare meristic data and were performed using PAST v3.12.

### DNA extraction, amplification, sequencing and alignment

Total DNA extraction was performed with the DNeasy Blood & Tissue Extraction Kit (Qiagen, Valencia, CA, USA), for 47 specimens of *Pareiorhaphis hystrix* plus five outgroup specimens (S1 Table) following the manufacturer's protocol. DNA amplification was performed by Polymerase Chain Reaction (PCR) for the following mitochondrial genes: *Cytochrome c oxidase subunit I* (*coI*), *Cytochrome b* (*cytb*), 16S, and 12S. These fragments were amplified using previously published and available primers (Table 2). The PCR protocol for *coI* included: initial denaturation at 94˚C for 2 min, followed by 40 cycles of denaturation at 94˚C for 20s (or 30s when using primers LCOI490 and HCO2198), annealing at 50˚C, 48˚C, 46˚C, 44˚C, 42˚C and 40˚C with 20s each end and 5s for intermediates, extension at 72˚C for 2 min and final extension at 72˚C for 10 min. For *cytb*: initial denaturation at 94˚C for 2 min, followed by 35 cycles of denaturation at 94˚C for 30s, annealing at 57˚C, 55˚C and 53˚C with 20s each, extension at 72˚C for 1.5 min and final extension at 72˚C for 10 min. For 16S: initial denaturation at 94˚C for 2 min, followed by 35 cycles of denaturation at 94˚C for 30s, annealing at 48 or 40˚C for 20s, extension at 72˚C for 60s and final extension at 72˚C for 10 min. Finally, for 12S: initial denaturation at 94˚C for 2 min, followed by 35 cycles of denaturation at 95˚C for 30s, annealing at 52˚C, 50˚C and 48˚C with 20s each, extension at 72˚C for 80s and final extension at 72˚C for 10 min. Negative controls were used in all procedures. The amplified DNA fragments were stained with Gelred and visualized on agarose gel. All plates with PCR products were shipped to Functional Bioscience Inc., United States, for purification and sequencing.

Geneious R8 6.0.5 [22] was used for editing and obtaining consensus sequences and automatic sequence alignment (*coI*, *cytb*), using the MUSCLE algorithm [23]. SATE v2 was employed for aligning 12S and 16S genes, also using MUSCLE. The sequences of the four mitochondrial genes were subsequently visually inspected and the alignments concatenated in Geneious.

**Table 2. Region and size of amplified genes, primer, and primer sequence.**

| Amplified region and size | Primer | Reference | Primer sequence |
|---|---|---|---|
| *coI* (447 bp) | LCOI490 | Hebert *et al.* [16] | 5'GGTCAACAAATCATAAAGATATTGG-3' |
| | HCO2198 | | 5'TAAACTTCAGGGTGACCAAAAAATCA-3' |
| | L6252 | Melo *et al.* [17] | 5'AAGGCGGGGAAAGCCCCGGCAG -3' |
| | H7271 | | 5'TCCTATGTAGCCGAATGGTTCTTTT-3' |
| | COCKTAILS: | Ward *et al.* [18]; Ivanova et al. [19] | 5'TGTAAAACGACGGCCAGTCAACCAACCACAAAGACATTGGCAC-3 |
| | VF2_t1 | | 5'TGTAAAACGACGGCCAGTCGACTAATCATAAGATATCGGCAC-3' |
| | Fish F2_t1 | | 5'CAGGAAACAGCTATGACACTTCAGGGTGCCGAAGAATCAGAA-3' |
| | Fish R2_t1 | | 5'CAGGAAACAGCTATGACACCTCAGGGTGTCC GAARAAYCARAA-3' |
| | FR1d_t1 | | |
| cytb (843 bp) | *cytb*Fa | Lujan *et al.* [20] | 5'TCCCACCCGGACTCTAACCGA-3' |
| | *cytb*Ra | | 5'CCGGATTACAAGACCGGCGCT-3' |
| 16S (453 bp) | 16Sar | Adapted from Lujan *et al.* [20] | 5'CGCCTGTTTATCAAAAACAT-3' |
| | 16Sbr | | 5'CCGGTCTGAACTCAGATCACGT-3' |
| 12S (775 bp) | Phe-L941 | Roxo *et al.* [21] | 5'AAA TCA AAG CAT AAC ACT GAA GAT G-3' |
| | Val-H2010 | | 5'CCA ATT TGC ATG GAT GTC TTC TCG G-3' |

## Molecular analyses

Genetic diversity parameters calculated were number of polymorphic sites (s), number of haplotypes (n), haplotype diversity (hd), and nucleotide diversity ($\pi$), using the software Arlequin v3.5 [24]. Network v5.3 was used to construct individual and concatenated haplotype networks applying the median-joining algorithm that identifies the most closely related haplotypes [25]. Concatenated genes, as well as the individual gene *coI*, were used for building a genetic distance matrix, calculated in MEGA v6 [26], using the Tamura Nei model (TN93), and respective standard errors were estimated with 10,000 bootstrap iterations.

The phylogenetic analysis was performed by Bayesian inference (BI). Partition Finder v1.1.1 [27] was used to selected models of nucleotide substitution for each mitochondrial gene, including separate codon positions for the codifying genes: *coI*, position one (JC), position two (F81), position three (TrN + G); *cytb*, position one (K80 + I), position two (HKY), position three (TrN + G); 16S (HKY + I) and 12S (TrNef). Subsequently, a Bayesian analysis was performed using MrBayes v3.2.6 [28] through the CIPRES supercomputing cluster (http://www.phylo.org/index.php; [29]). MrBayes was set to run 100 million generations using four chains (nchain = 4), two parallel runs, sampling every 1000 trees with the first 25% trees discarded as burn in. The outgroup was composed of *Pareiorhaphis azygolechis*, *P. steindachneri* (Miranda Ribeiro, 1918), and *P. vestigipinnis* (Pereira & Reis, 1992).

Delimitation of possible multiple species was also tested using the *General Mixed Yule Coalescent* (GMYC) model, available on the GMYC server (https://species.h-its.org/gmyc; [30]). The coalescence trees were produced by Bayesian inference for the gene *coI*. The HKY model was used for all positions, considering relaxed molecular clock using a lognormal time distribution and birth-death prior through the softwares BEAUTi and BEAST. BEAST was programmed to run 100 million generations, sampling every 1,000 trees. TreeAnnotator [31] was used to build a consensus tree, with 10% of the trees discarded. To perform the analysis on the online server, the tree generated in TreeAnnotator was converted to Newick format using the software R v3 [32].

## Results

### Morphological observations

**Hypertrophied odontodes.** Several development patterns of cheek hyperthrophied odontodes and associated lateral fleshy lobe were observed in males. These odontodes vary in size, thickness, density, direction, and position on the sides of the head, while the fleshy lobe varies in thickness and shape (Fig 4). Such variation, however, occurs independently of the geographic area and is found concurrently within each area, not being useful to distinguish populations. Conversely, the odontodes on the pectoral-fin spine were more delicate and homogeneous than the pattern detected for cheek odontodes, without significant variation. However, it is possible to observe syntopic adult males with small and barely visible odontodes on the pectoral-fin spine (Fig 5A), and others with slightly larger odontodes (Fig 5B).

**Coloration.** The comparative analysis evidenced subtle differences in color pattern among individuals between and within areas, without a detectable geographical pattern. In general, the coloration varied from gray to light brown shades dorsally, with many scattered spots, and the ventral region usually with shades of pale yellow. Some specimens possess dark small or medium-sized dots clearly visible along the head and dorsal surface of the trunk (Fig 6A and 6B). Other specimens show a darker dorsal coloration, with a more mottled pattern of fine to coarse dark vermiculations (Fig 6C and 6D).

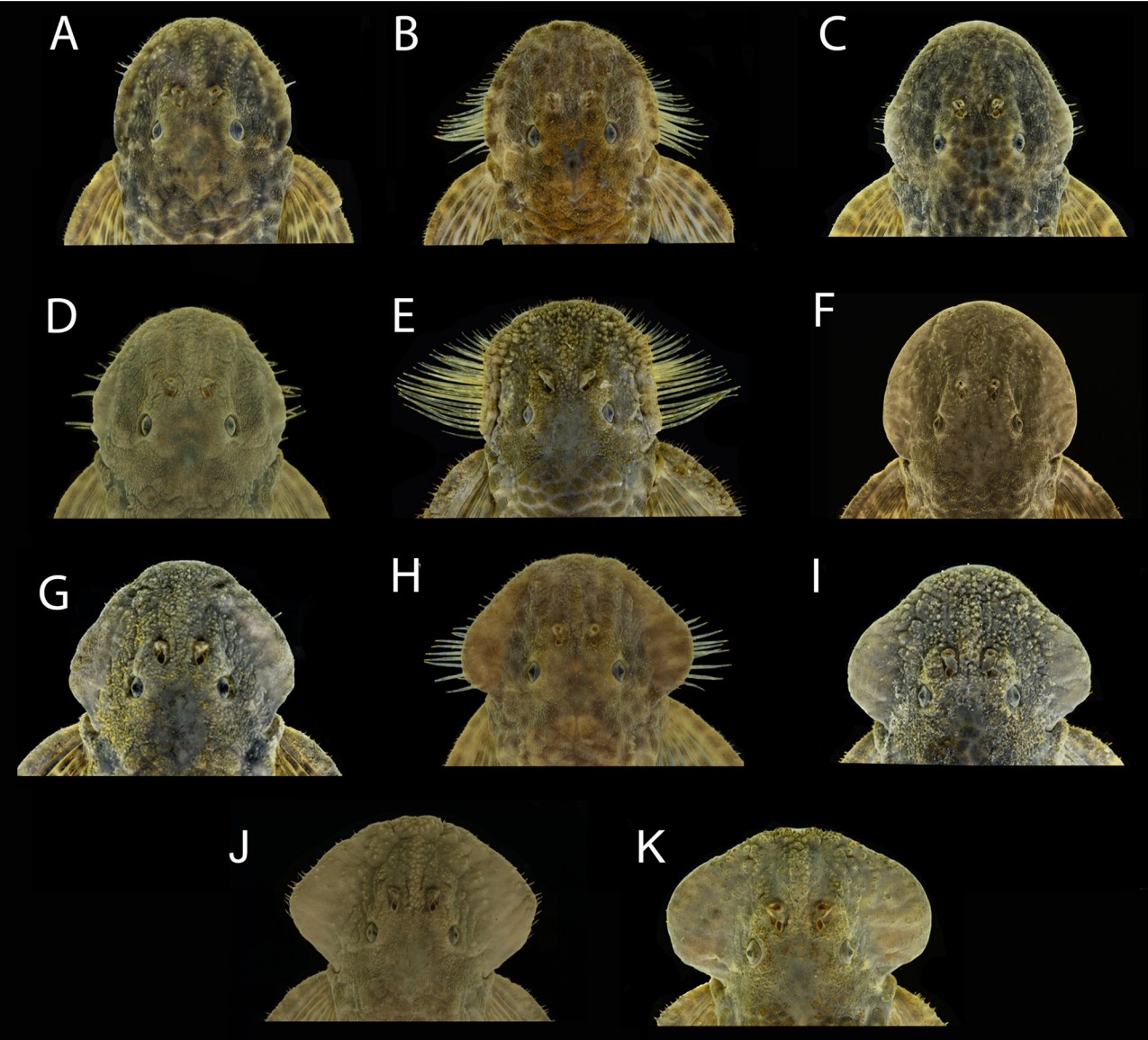

**Fig 4. Variation in development of cheek fleshy lobe and associated odontodes of males of *Pareiorhaphis hystrix*.** (A) Chapecó, MCP 40150; (B) Ijuí, MCP 11704; (C) Middle Uruguay, MCP 50946; (D) Passo Fundo, MCP 53257; (E) Middle Antas, MCP 48327; (F) Canoas, MCP 53259; (G) Prata, MCP 22787; (H) Ijuí, MCP 21191; (I) Upper Antas, MCP 50156; (J) Middle Antas, MCP 43518; (K) Upper Antas, UFRGS 8910.

**Abdominal plates.** Small, granular plates were observed in specimens from some of the study areas. Specimens from the Uruguay River basin showed a predominance of anterior abdominal plates, with the exception of those from Pelotas. Specimens from that area and those from areas in the Taquari River basin lack visible plates in the abdomen. The anterior abdominal plates are small and bear minute odontodes. They are located in the middle or at the edges of the pectoral girdle, usually just posterior to gills slits. In general, anterior abdominal plates occur only in adult specimens, both male and female (Fig 7A to 7E). One individual from Passo Fundo (Fig 7A) had a large number of abdominal plates on the pectoral girdle.

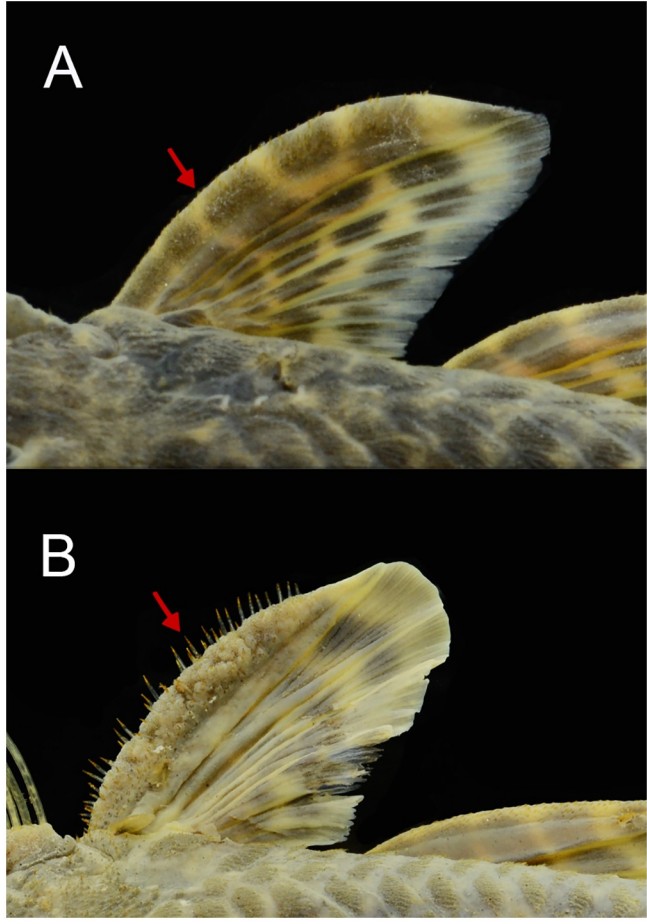

**Fig 5. Variation in development of pectoral-fin spine odontodes of males of *Pareiorhaphis hystrix*.** (A) short, Middle Uruguay, MCP 50946; (B) long, Middle Antas, MCP 48327. Arrows indicate the odontodes.

## Morphological analyses

Counts and measurements obtained from each of the nine populations/areas are shown in Tables 3 and 4. When comparing meristic data between the nine areas studied statistically significant variation was observed in the number of premaxillary teeth (ANOVA, p = 0.0003, f = 3.90), where Tukey's test indicates significant differentiation values between Canoas and the other locations (p <0.05), except Chapecó (p = 0.06), Ijuí (p = 0.69), and Middle Uruguay (p = 0.15) (Fig 8A to 8N); number of dentary teeth (ANOVA, p = 1.053E-07, f = 6.64), in which specimens from Canoas differed from those from other areas (p <0.05), except Ijuí (p = 0.7), and Pelotas differed significantly from Ijuí (p = 0.002).

The ANOVA also indicated significant variation in the number of lateral plates: in the median lateral series (p = 8.667e-13, f = 10.94), with Canoas differing from all other areas (p <0.05) except Chapecó (p = 0.8) and Passo Fundo (p = 0.6). In general, the areas of the Uruguay River (Chapecó, Ijuí, Passo Fundo, Canoas) differed from the Taquari River–Upper and Middle Antas (p <0.05), Pelotas differed from Chapecó (p = 0.03), Passo Fundo (p = 0.04), and Canoas (p = 1.218e-6). Concerning number of plates at the base of dorsal fin (p = 4.481e-8, f = 28.07), specimens from the Uruguay River basin (except Pelotas; p >0.05) generally differed statistically from those in the Taquari River basin (p <0.05). Plates between dorsal and

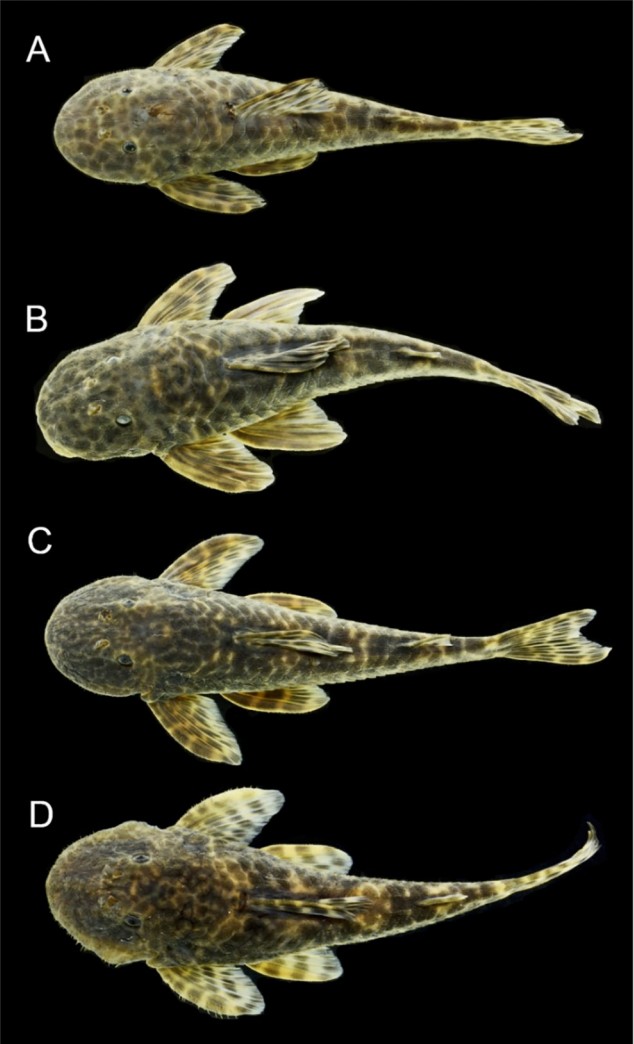

**Fig 6. Variation in color pattern of *Pareiorhaphis hystrix*.** (A) small dark blotches, female, Ijuí, MCP 48639, 75.3 mm SL; (B) large dark blotches, female, Ijuí, MCP 44995, 89.7 mm SL; (C) coarse vermiculations, male, Middle Uruguay, MCP 51452, 86.4 mm SL; (D) fine vermiculations, male, Upper Antas, UFRGS 21850, 82.9 mm SL.

adipose fins (p = 4.248e-11, f = 9.4), where specimens of the Uruguay River basin differ statistically from the Taquari River (Ijuí vs. Upper Antas, Passo Fundo vs. Upper and Middle Antas, Canoas vs. Upper Antas; p <0.05), except for Pelotas. Plates between anal and caudal fins (p = 7.818e-6, f = 5.13), where Passo Fundo differs significantly from most areas (Upper and Middle Antas, Chapecó, Pelotas, Middle Uruguay, and Canoas), and Pelotas showed greater proximity to the Taquari River basin.

ANOVA detected no significant variation between the nine study areas for the following meristic: plates between adipose and caudal fins (p = 0.07, f = 1.83); plates at the base of anal fin (p >0.05); pre-adipose azygous plates (no variance); number of branched dorsal-fin rays (p >0.05); number of branched pectoral-fin rays (no variance); number of branched pelvic-fin rays (no variance); number of branched anal-fin rays (p >0.05), and number of branched caudal-fin rays (p >0.05).

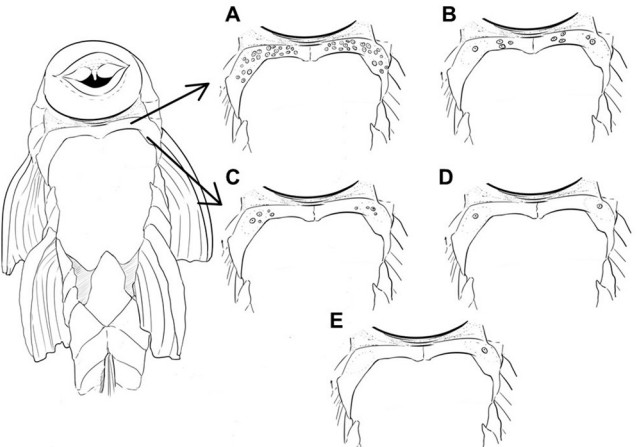

**Fig 7. Schematic drawing of anterior abdominal plate variation in specimens of *Pareiorhaphis hystrix*.** (A) Passo Fundo; (B) Chapecó; (C) Middle Uruguay; (D) Canoas; (E) Ijuí.

The principal component analysis (PCA) did not differentiate the nine areas, which were generally overlapping. However, it evidences a slight separation by river basin, except for Prata and Pelotas, which were closer to their respective adjacent basins (Fig 9). Principal Component 1 (PC1) accounted for 28.9% of the total variance, PC2 for 20.1% and PC3 for 7.8%.

The Linear discriminant analysis (LDA) (Fig 10) revealed better differentiation between basins in comparison to PCA, with Pelotas clustering with areas of the Taquari basin. The percent separation achieved by each discriminant function was 51.8% for LDA1, 20.2% for LDA2, 10.3% for LDA3 and 6.5% for LDA4. The loads for discriminant function LDA1 suggest that the most significant measures were: (1) caudal-peduncle length (0.22), for which specimens from Middle Antas, Middle Uruguay, Ijuí, and Canoas presented higher length, with Middle Antas also grouping the shortest lengths along with Pelotas. (2) Post-dorsal length (0.19), where the areas of the Uruguay River basin presented the highest values. (3) Head depth (-0.32), with specimens from the Taquari basin—specifically those from the Middle and Upper Antas, and from Pelotas (Uruguay River basin) with the highest values. (4) Body depth at origin of dorsal fin (-0.5), with specimens from the Uruguay River basin—except Pelotas, showing shallower body.

For LDA2, the following measurements were the most significant: (1) anal-fin spine length (0.26), with specimens from Upper Antas, Middle Antas, and Pelotas showing the highest values, while Ijuí had the lowest. (2) Preadipose length (0.22), with specimens from Upper Antas having highest value, despite the minimal variation between areas. (3) Body width at origin of dorsal fin (-0.19), with Chapecó specimens showing the narrowest bodies, while those of Upper Antas, Middle Antas, and Pelotas having the widest. (4) Adipose-fin spine length (-0.26), with the longest spine for specimens from Prata, although the average between the areas are very close. In general, the variation observed between the areas by the LDA were minimal, with very close values.

## Molecular data

An alignment of 2,518 base pairs (bp) was obtained for the mitochondrial genes *coI* (447 bp), *cytb* (843 bp), 16S (453 bp), and 12S (775 bp). Nucleotide composition is as follows: Cytosine (C): 28.45%; Thymine (T): 25.09%; Adenine (A): 26.89%, and Guanine (G): 19.55%. Higher haplotype diversity (hd) compared to nucleotide diversity (π) was observed for the four genes

**Table 3. Morphometric and meristic data of *Pareiorhaphis hystrix*.**

| | Upper Antas n = 34 | | | | Middle Antas n = 25 | | | | Prata n = 25 | | | | Pelotas n = 25 | | | |
|---|---|---|---|---|---|---|---|---|---|---|---|---|---|---|---|---|
| | Min | Max | Mean | SD | Min | Max | Mean | SD | Min | Max | Mean | SD | Min | Max | Mean | SD |
| Standard length (mm) | 65.4 | 99.9 | 80.9 | | 70.1 | 103.5 | 80.1 | | 66.5 | 99.2 | 80.2 | | 70.0 | 98.2 | 80.2 | |
| **Percent of standard length** | | | | | | | | | | | | | | | | |
| Head length | 29.9 | 34.1 | 32.3 | 0.9 | 30.7 | 35.4 | 32.3 | 1.3 | 30.1 | 35.7 | 32.5 | 1.4 | 30.6 | 35.4 | 32.6 | 1.1 |
| Trunk length | 14.0 | 18.8 | 16.2 | 1.1 | 14.5 | 18.7 | 16.4 | 1.0 | 14.0 | 17.3 | 15.8 | 0.9 | 14.7 | 18.4 | 16.2 | 1.0 |
| Abdominal length | 23.3 | 28.7 | 26.1 | 1.1 | 23.6 | 28.2 | 26.1 | 1.4 | 24.9 | 29.3 | 26.5 | 1.1 | 23.5 | 28.5 | 26.3 | 1.3 |
| Predorsal length | 42.6 | 46.2 | 44.0 | 0.8 | 42.1 | 46.5 | 44.2 | 1.2 | 41.0 | 45.3 | 43.3 | 0.9 | 43.0 | 47.0 | 44.9 | 1.2 |
| Dorsal-fin spine length | 19.9 | 27.3 | 22.6 | 1.4 | 19.4 | 24.9 | 21.7 | 1.1 | 19.2 | 25.1 | 21.6 | 1.4 | 19.4 | 25.5 | 22.7 | 1.4 |
| Body depth at dorsal origin | 17.4 | 22.2 | 19.6 | 1.2 | 15.1 | 22.7 | 19.0 | 1.6 | 16.0 | 20.8 | 18.3 | 1.1 | 17.0 | 22.3 | 19.1 | 1.5 |
| Body width at dorsal origin | 22.4 | 28.0 | 24.3 | 1.1 | 22.5 | 29.0 | 24.9 | 1.4 | 22.7 | 27.3 | 24.5 | 1.1 | 21.9 | 28.5 | 24.2 | 1.8 |
| Posdorsal length | 38.6 | 45.0 | 41.3 | 1.5 | 36.5 | 44.3 | 41.1 | 2.1 | 38.8 | 44.8 | 41.0 | 1.4 | 36.7 | 46.6 | 40.6 | 2.1 |
| Pre-adipose fin length | 78.4 | 84.1 | 81.0 | 1.2 | 77.7 | 83.0 | 81.0 | 1.2 | 77.3 | 81.7 | 79.7 | 1.2 | 77.5 | 83.7 | 80.9 | 1.5 |
| Adipose-fin spine length | 7.9 | 11.2 | 9.4 | 0.8 | 7.4 | 11.3 | 9.1 | 0.9 | 8.3 | 16.4 | 10.0 | 1.5 | 6.6 | 9.9 | 8.3 | 0.8 |
| Distance adipose to caudal fin | 16.2 | 21.3 | 18.9 | 1.1 | 16.8 | 21.5 | 19.0 | 1.1 | 17.3 | 21.3 | 19.2 | 1.1 | 16.0 | 21.5 | 18.5 | 1.2 |
| Preanal length | 62.1 | 66.9 | 63.8 | 1.3 | 60.7 | 67.1 | 64.4 | 1.9 | 60.3 | 66.0 | 63.1 | 1.5 | 60.3 | 66.6 | 63.9 | 1.5 |
| Anal-fin spine length | 14.0 | 19.8 | 16.6 | 1.5 | 13.3 | 20.5 | 15.8 | 1.5 | 12.8 | 17.3 | 14.6 | 1.2 | 12.2 | 20.5 | 16.6 | 1.9 |
| Pectoral-fin spine length | 20.1 | 27.9 | 25.2 | 1.6 | 22.8 | 28.7 | 25.2 | 1.4 | 19.9 | 26.7 | 24.0 | 1.5 | 21.5 | 28.0 | 24.7 | 1.5 |
| Pelvic-fin spine length | 19.9 | 27.8 | 23.6 | 2.0 | 21.5 | 28.5 | 23.8 | 1.9 | 21.6 | 26.5 | 23.5 | 1.4 | 20.4 | 28.5 | 23.7 | 2.1 |
| Cleithral width | 28.1 | 32.3 | 30.5 | 1.0 | 27.7 | 34.7 | 30.5 | 1.4 | 29.0 | 32.8 | 30.9 | 1.0 | 28.3 | 32.6 | 30.7 | 1.1 |
| Upper caudal-fin ray | 20.5 | 27.3 | 24.2 | 1.6 | 20.5 | 27.0 | 23.4 | 1.7 | 20.4 | 25.3 | 22.9 | 1.3 | 20.1 | 26.6 | 23.1 | 1.7 |
| Lower caudal-fin ray | 21.8 | 30.1 | 26.8 | 1.6 | 23.0 | 31.0 | 26.1 | 1.9 | 22.3 | 29.9 | 26.2 | 1.8 | 22.4 | 29.5 | 26.2 | 2.0 |
| Body width at anal origin | 11.8 | 16.7 | 14.8 | 1.2 | 12.9 | 16.8 | 14.7 | 1.0 | 12.6 | 15.9 | 14.5 | 0.8 | 11.8 | 15.7 | 13.8 | 0.9 |
| Caudal peduncle length | 33.2 | 37.9 | 35.3 | 1.3 | 31.8 | 39.0 | 35.0 | 1.8 | 33.5 | 38.0 | 35.6 | 1.1 | 31.9 | 38.0 | 35.0 | 1.6 |
| Caudal peduncle depth | 8.4 | 10.2 | 9.1 | 0.4 | 8.6 | 10.0 | 9.0 | 0.3 | 8.2 | 10.5 | 9.1 | 0.5 | 7.4 | 9.6 | 8.7 | 0.7 |
| Caudal peduncle width | 4.4 | 6.9 | 6.0 | 0.7 | 4.4 | 7.2 | 5.7 | 0.7 | 4.4 | 6.6 | 5.4 | 0.5 | 4.4 | 8.5 | 6.0 | 0.9 |
| **Percent of head length** | | | | | | | | | | | | | | | | |
| Snout length | 56.0 | 68.3 | 62.0 | 2.6 | 56.4 | 67.4 | 62.0 | 2.3 | 58.3 | 65.7 | 62.2 | 1.8 | 59.2 | 65.7 | 62.3 | 1.6 |
| Orbital diameter | 10.9 | 14.8 | 12.7 | 1.0 | 8.8 | 14.7 | 12.5 | 1.2 | 10.0 | 13.8 | 12.4 | 1.0 | 10.5 | 15.5 | 12.6 | 1.2 |
| Least interorbital width | 32.4 | 39.6 | 35.1 | 1.5 | 30.6 | 39.1 | 35.3 | 2.2 | 31.1 | 38.3 | 35.2 | 1.9 | 32.2 | 36.6 | 34.4 | 1.2 |
| Head depth | 45.2 | 57.1 | 51.3 | 2.3 | 47.8 | 61.6 | 51.1 | 2.9 | 45.7 | 55.0 | 49.0 | 2.4 | 46.4 | 61.1 | 52.1 | 3.2 |
| **Counts** | | | | | | | | | | | | | | | | |
| Premaxillary teeth | 38 | 54 | 45.4 | 3.7 | 39 | 51 | 44.4 | 3.0 | 40 | 59 | 45.7 | 4.9 | 33 | 59 | 45.7 | 6.0 |
| Dentary teeth | 35 | 57 | 45.1 | 4.3 | 41 | 51 | 45.3 | 3.2 | 40 | 55 | 46.0 | 4.8 | 35 | 62 | 46.4 | 5.7 |
| Median series lateral plates | 27 | 29 | 27.4 | 0.5 | 27 | 29 | 27.3 | 0.6 | 27 | 28 | 27.2 | 0.4 | 26 | 30 | 27.3 | 0.7 |
| Plates at base of dorsal-fin | 6 | 8 | 7.1 | 0.4 | 6 | 8 | 7.5 | 0.6 | 7 | 8 | 7.4 | 0.5 | 6 | 9 | 7.2 | 0.6 |
| Plates between dorsal/adipose | 7 | 9 | 8.5 | 0.6 | 7 | 9 | 8.0 | 0.5 | 6 | 9 | 7.7 | 0.6 | 8 | 9 | 8.5 | 0.5 |
| Plates between adipose/caudal | 3 | 4 | 3.8 | 0.4 | 3 | 5 | 4.1 | 0.4 | 4 | 5 | 4.0 | 0.2 | 3 | 5 | 4.0 | 0.5 |
| Plates lateral to anal fin | 3 | 4 | 3.3 | 0.5 | 3 | 4 | 3.2 | 0.4 | 3 | 4 | 3.0 | 0.2 | 3 | 4 | 3.4 | 0.5 |
| Plates between anal/caudal | 13 | 14 | 13.1 | 0.4 | 12 | 13 | 12.9 | 0.3 | 12 | 13 | 12.8 | 0.4 | 12 | 14 | 13.2 | 0.6 |
| Azygous pre-adipose plates | 1 | 1 | 1.0 | 0.0 | 0 | 1 | 1.0 | 0.2 | 1 | 1 | 1.0 | 0.0 | 1 | 1 | 1.0 | 0.0 |
| Dorsal-fin branched rays | 7 | 8 | 7.1 | 0.3 | 7 | 7 | 7.0 | 0.0 | 7 | 8 | 7.0 | 0.2 | 7 | 7 | 7.0 | 0.0 |
| Pectoral-fin branched rays | 6 | 6 | 6.0 | 0.0 | 5 | 6 | 6.0 | 0.2 | 6 | 6 | 6.0 | 0.0 | 5 | 6 | 6.0 | 0.2 |
| Pelvic-fin branched rays | 5 | 5 | 5.0 | 0.0 | 5 | 5 | 5.0 | 0.0 | 5 | 5 | 5.0 | 0.0 | 5 | 5 | 5.0 | 0.0 |
| Anal-fin branched rays | 4 | 5 | 5.0 | 0.2 | 5 | 5 | 5.0 | 0.0 | 4 | 5 | 5.0 | 0.2 | 5 | 5 | 5.0 | 0.0 |
| Caudal-fin branched rays | 13 | 14 | 14.0 | 0.2 | 12 | 14 | 13.9 | 0.4 | 13 | 15 | 14.0 | 0.3 | 13 | 14 | 14.0 | 0.2 |

n = number of specimens; SD = standard deviation.

**Table 4. Morphometric and meristic data of *Pareiorhaphis hystrix*.**

| | Chapecó n = 14 | | | | Ijuí n = 26 | | | | Passo Fundo n = 16 | | | | Middle Uruguay n = 22 | | | | Canoas n = 24 | | | |
|---|---|---|---|---|---|---|---|---|---|---|---|---|---|---|---|---|---|---|---|---|
| | Min | Max | Mean | SD | Min | Max | Mean | SD | Min | Max | Mean | SD | Min | Max | Mean | SD | Min | Max | Mean | SD |
| Standard length (mm) | 60.5 | 92.6 | 76.8 | | 75.3 | 109.3 | 85.3 | | 61.8 | 84.8 | 72.2 | | 70.0 | 95.9 | 81.2 | | 72.1 | 114.7 | 89.6 | |
| **Percent of stand length** | | | | | | | | | | | | | | | | | | | | |
| Head length | 30.5 | 34.3 | 32.2 | 1.3 | 28.9 | 33.5 | 31.8 | 1.3 | 29.7 | 33.6 | 32.4 | 1.0 | 29.8 | 34.3 | 31.8 | 1.1 | 30.6 | 33.9 | 32.3 | 0.9 |
| Trunk length | 13.0 | 16.7 | 15.4 | 1.1 | 12.3 | 17.1 | 15.6 | 1.0 | 14.9 | 17.2 | 15.9 | 0.7 | 14.0 | 16.7 | 15.3 | 0.8 | 14.5 | 17.1 | 15.6 | 0.7 |
| Abdominal length | 23.9 | 28.4 | 25.7 | 1.2 | 23.8 | 28.3 | 25.6 | 1.2 | 23.2 | 27.7 | 25.6 | 1.2 | 24.0 | 27.3 | 25.6 | 0.8 | 24.1 | 27.9 | 26.2 | 1.0 |
| Predorsal length | 40.0 | 45.2 | 43.4 | 1.7 | 40.1 | 44.9 | 42.9 | 1.5 | 40.3 | 44.7 | 43.0 | 1.1 | 41.8 | 45.7 | 43.1 | 1.0 | 41.2 | 45.0 | 43.3 | 0.9 |
| Dorsal-fin spine length | 19.4 | 23.7 | 21.8 | 1.3 | 20.0 | 23.7 | 21.7 | 1.1 | 19.4 | 25.7 | 22.8 | 1.7 | 19.6 | 23.2 | 21.3 | 0.9 | 21.3 | 24.4 | 22.8 | 0.9 |
| Body depth at dorsal origin | 14.5 | 18.3 | 16.4 | 1.0 | 13.7 | 18.6 | 16.4 | 1.1 | 14.1 | 19.7 | 16.8 | 1.5 | 13.6 | 19.2 | 16.1 | 1.4 | 14.6 | 20.2 | 17.5 | 1.3 |
| Body width at dorsal origin | 20.1 | 23.9 | 22.2 | 1.2 | 20.6 | 25.7 | 22.7 | 1.2 | 19.6 | 27.0 | 22.7 | 1.9 | 19.7 | 25.6 | 22.5 | 1.3 | 22.1 | 27.3 | 24.7 | 1.3 |
| Posdorsal length | 39.8 | 44.9 | 42.4 | 1.4 | 39.3 | 45.3 | 42.3 | 1.5 | 39.8 | 44.6 | 42.7 | 1.2 | 39.1 | 46.7 | 42.0 | 1.5 | 36.9 | 44.9 | 41.8 | 1.6 |
| Pre-adipose fin length | 77.3 | 83.5 | 80.3 | 1.8 | 77.4 | 82.1 | 79.8 | 1.2 | 77.1 | 81.2 | 78.8 | 1.3 | 77.2 | 83.6 | 80.5 | 1.4 | 77.8 | 81.8 | 79.6 | 1.0 |
| Adipose-fin spine length | 7.4 | 9.3 | 8.3 | 0.6 | 7.5 | 10.2 | 8.9 | 0.8 | 7.6 | 9.6 | 8.4 | 0.6 | 7.8 | 11.1 | 8.9 | 0.7 | 7.1 | 11.1 | 9.0 | 0.9 |
| Dist adipose to caudal fin | 15.6 | 21.7 | 19.5 | 1.8 | 17.4 | 23.1 | 20.1 | 1.5 | 18.6 | 22.6 | 20.8 | 1.0 | 17.3 | 22.5 | 19.6 | 1.3 | 16.9 | 21.3 | 19.5 | 0.9 |
| Preanal length | 61.1 | 66.6 | 63.6 | 1.6 | 60.6 | 66.1 | 63.0 | 1.4 | 60.3 | 65.5 | 62.5 | 1.5 | 60.5 | 65.2 | 62.9 | 1.1 | 59.9 | 65.2 | 62.7 | 1.4 |
| Anal-fin spine length | 13.1 | 17.2 | 15.0 | 1.2 | 11.9 | 17.4 | 15.3 | 1.4 | 13.8 | 17.5 | 15.7 | 1.1 | 13.0 | 17.6 | 14.9 | 1.1 | 13.4 | 16.9 | 15.4 | 1.0 |
| Pectoral-fin spine length | 21.4 | 26.0 | 23.8 | 1.1 | 22.0 | 28.0 | 25.3 | 1.5 | 23.0 | 27.1 | 24.7 | 1.3 | 22.8 | 27.4 | 24.7 | 1.3 | 23.0 | 27.5 | 26.1 | 1.2 |
| Pelvic-fin spine length | 21.7 | 26.2 | 24.0 | 1.4 | 21.2 | 27.1 | 24.1 | 1.3 | 23.0 | 27.1 | 24.5 | 1.1 | 22.2 | 26.1 | 24.1 | 1.0 | 23.3 | 26.7 | 25.2 | 1.0 |
| Cleithral width | 27.4 | 31.8 | 29.7 | 1.3 | 26.7 | 31.8 | 29.5 | 1.3 | 27.9 | 31.9 | 29.8 | 1.2 | 26.8 | 30.8 | 28.4 | 1.1 | 27.8 | 32.5 | 30.8 | 1.1 |
| Upper caudal-fin ray | 21.8 | 25.4 | 23.3 | 1.3 | 20.6 | 24.9 | 22.6 | 1.1 | 21.0 | 26.9 | 23.1 | 1.7 | 19.6 | 25.5 | 22.7 | 1.5 | 22.4 | 25.9 | 24.1 | 0.9 |
| Lower caudal-fin ray | 22.7 | 28.7 | 26.0 | 1.9 | 22.3 | 27.9 | 25.6 | 1.4 | 22.6 | 29.9 | 26.5 | 2.0 | 23.9 | 29.0 | 25.9 | 1.6 | 22.0 | 29.7 | 27.7 | 1.6 |
| Body width at anal origin | 10.6 | 13.2 | 12.2 | 0.8 | 11.8 | 15.4 | 13.6 | 0.9 | 10.4 | 17.5 | 13.5 | 1.5 | 12.5 | 15.5 | 13.9 | 0.8 | 12.9 | 15.4 | 14.2 | 0.6 |
| Caudal peduncle length | 33.8 | 38.7 | 36.1 | 1.4 | 34.0 | 39.8 | 36.6 | 1.3 | 34.0 | 38.6 | 36.1 | 1.4 | 32.9 | 39.7 | 36.8 | 1.4 | 33.4 | 39.1 | 36.2 | 1.3 |
| Caudal peduncle depth | 6.9 | 8.8 | 7.9 | 0.6 | 6.8 | 9.1 | 8.0 | 0.4 | 7.7 | 9.0 | 8.5 | 0.4 | 6.8 | 8.3 | 7.5 | 0.4 | 7.4 | 8.5 | 7.9 | 0.3 |
| Caudal peduncle width | 4.0 | 5.9 | 5.0 | 0.5 | 4.6 | 6.2 | 5.2 | 0.4 | 4.3 | 6.1 | 5.5 | 0.5 | 4.3 | 6.1 | 5.0 | 0.5 | 4.8 | 6.2 | 5.5 | 0.4 |
| **Percent of head length** | | | | | | | | | | | | | | | | | | | | |
| Snout length | 56.9 | 65.1 | 62.4 | 1.9 | 61.0 | 65.6 | 63.4 | 1.3 | 62.6 | 66.8 | 64.3 | 1.2 | 58.5 | 68.0 | 64.6 | 2.1 | 60.5 | 67.6 | 64.2 | 1.6 |
| Orbital diameter | 9.7 | 14.9 | 12.5 | 1.3 | 9.3 | 14.6 | 12.7 | 1.3 | 9.4 | 13.8 | 11.7 | 1.4 | 10.2 | 13.8 | 11.7 | 0.8 | 10.4 | 14.2 | 12.4 | 0.9 |
| Least interorbital width | 30.5 | 35.3 | 33.3 | 1.5 | 30.2 | 36.9 | 33.9 | 1.5 | 31.5 | 36.8 | 34.2 | 1.3 | 31.1 | 37.0 | 33.9 | 1.5 | 31.5 | 36.3 | 34.3 | 1.4 |
| Head depth | 42.8 | 51.4 | 46.2 | 2.4 | 43.2 | 50.1 | 46.8 | 1.7 | 43.2 | 51.1 | 46.8 | 2.3 | 41.5 | 49.8 | 45.8 | 2.5 | 44.9 | 50.3 | 47.5 | 1.4 |
| **Counts** | | | | | | | | | | | | | | | | | | | | |
| Premaxillary teeth | 38 | 52 | 45.4 | 4.0 | 37 | 48 | 42.7 | 3.0 | 35 | 56 | 45.1 | 5.5 | 38 | 51 | 44.0 | 3.6 | 36 | 47 | 40.6 | 3.4 |
| Dentary teeth | 36 | 51 | 44.9 | 4.0 | 37 | 48 | 41.9 | 2.7 | 39 | 54 | 44.6 | 4.8 | 38 | 48 | 43.9 | 2.7 | 36 | 47 | 39.9 | 2.7 |
| Median series lateral plates | 26 | 27 | 26.7 | 0.5 | 26 | 28 | 26.8 | 0.5 | 26 | 28 | 26.7 | 0.6 | 26 | 29 | 27.1 | 0.6 | 25 | 27 | 26.3 | 0.6 |
| Plates at base of dorsal-fin | 5 | 7 | 6.2 | 0.6 | 6 | 7 | 6.6 | 0.5 | 5 | 7 | 6.2 | 0.5 | 5 | 7 | 6.2 | 0.5 | 5 | 7 | 6.2 | 0.5 |
| Plates betwn dorsal/adipose | 8 | 9 | 8.3 | 0.5 | 7 | 9 | 7.8 | 0.5 | 7 | 8 | 7.4 | 0.5 | 7 | 9 | 8.0 | 0.7 | 7 | 9 | 7.8 | 0.5 |
| Plates between adip/caudal | 3 | 5 | 3.9 | 0.5 | 3 | 4 | 3.9 | 0.3 | 3 | 5 | 3.9 | 0.6 | 3 | 5 | 3.7 | 0.5 | 3 | 4 | 3.8 | 0.4 |
| Plates lateral to anal fin | 3 | 4 | 3.1 | 0.3 | 3 | 4 | 3.1 | 0.3 | 3 | 3 | 3.0 | 0.0 | 3 | 3 | 3.0 | 0.0 | 3 | 3 | 3.0 | 0.0 |
| Plates between anal/caudal | 12 | 14 | 13.1 | 0.5 | 11 | 13 | 12.6 | 0.6 | 12 | 13 | 12.3 | 0.5 | 11 | 14 | 12.9 | 1.0 | 12 | 13 | 12.9 | 0.3 |
| Azygous pre-adipose plates | 1 | 1 | 1.0 | 0.0 | 1 | 1 | 1 | 0.0 | 1 | 1 | 1.0 | 0 | 1 | 1 | 1.0 | 0.0 | 0 | 1 | 0.9 | 0.3 |
| Dorsal-fin branched rays | 7 | 7 | 7.0 | 0.0 | 7 | 7 | 7.0 | 0 | 7 | 7 | 7.0 | 0 | 7 | 7 | 7.0 | 0.0 | 7 | 7 | 7.0 | 0 |
| Pectoral-fin branched rays | 5 | 6 | 5.9 | 0.3 | 6 | 6 | 6.0 | 0 | 6 | 6 | 6.0 | 0 | 6 | 6 | 6.0 | 0.0 | 6 | 6 | 6.0 | 0 |
| Pelvic-fin branched rays | 5 | 5 | 5.0 | 0.0 | 5 | 5 | 5.0 | 0 | 5 | 5 | 5.0 | 0 | 5 | 5 | 5.0 | 0.0 | 5 | 5 | 5.0 | 0 |
| Anal-fin branched rays | 5 | 5 | 5.0 | 0.0 | 5 | 5 | 5.0 | 0 | 4 | 5 | 4.9 | 0.2 | 5 | 5 | 5.0 | 0.0 | 5 | 5 | 5.0 | 0 |
| Caudal-fin branched rays | 14 | 14 | 14.0 | 0.0 | 13 | 14 | 14.0 | 0.2 | 13 | 14 | 13.9 | 0.2 | 14 | 15 | 14.0 | 0.2 | 12 | 14 | 13.9 | 0.4 |

n = number of specimens; SD = standard deviation.

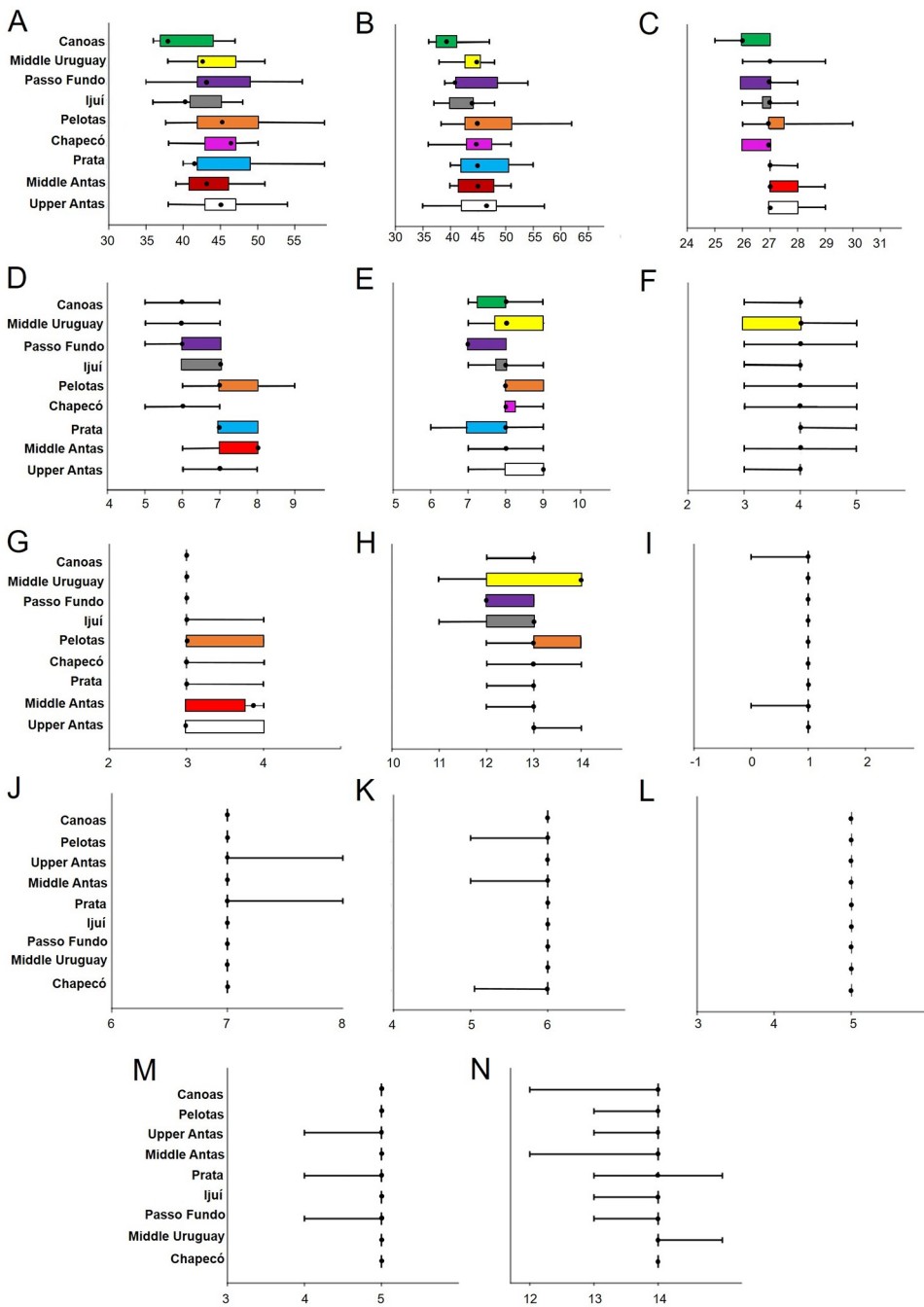

**Fig 8. Box plots of variable meristic data between different areas of occurrence of *Pareiorhaphis hystrix* according to Tukey's Pairwise results.** Each graphic contains the number of: (A) premaxillary teeth, left side; (B) dentary teeth, left side; (C) median lateral plates; (D) plates at dorsal-fin base; (E) plates between dorsal and adipose fins; (F) plates between adipose and caudal fins; (G) plates at anal-fin base; (H) plates between anal and caudal fins; (I) pre-adipose azygous plates; (J) branched dorsal-fin rays; (K) branched pectoral-fin rays; (L) branched pelvic-fin rays; (M) branched anal-fin rays; and (N) branched caudal-fin rays. For each sample, the 25 to 75% quartiles are indicated by a box; the mode is represented by a dot; the minimum and maximum values are shown as short vertical lines.

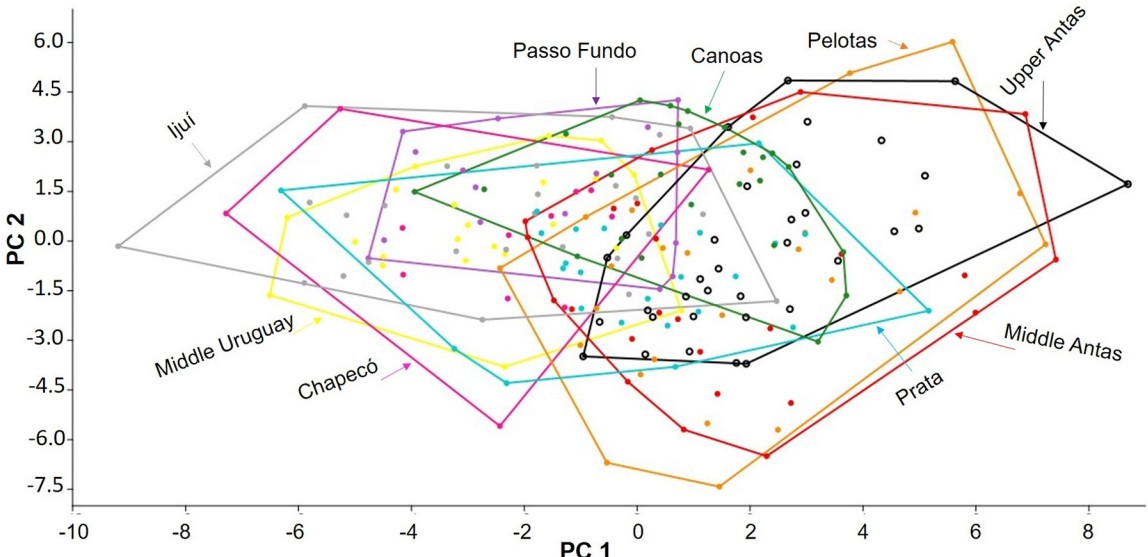

**Fig 9. Scatter plot of principal components PC1 x PC2 factor scores.** Each color represents an area studied in the Taquari River and Uruguay River basins.

(Table 5). For *coI*, the area with highest haplotypic diversity was Prata (1.00 +/- 0.50). For *cytb*, haplotypic diversity values were generally high, especially for Chapecó (0.90 +/- 0.16). The 16S had median diversity values, the highest for Middle Uruguay (0.60 +/- 0.17); and for the 12S, the highest diversity of haplotypes was that of Prata (1.00 +/- 0.27).

## Haplotype networks

The concatenated haplotype network (Fig 11) presented 22 haplotypes and one subtle genetic structure by geographic area, with unique and non-shared haplotypes. Pelotas presented

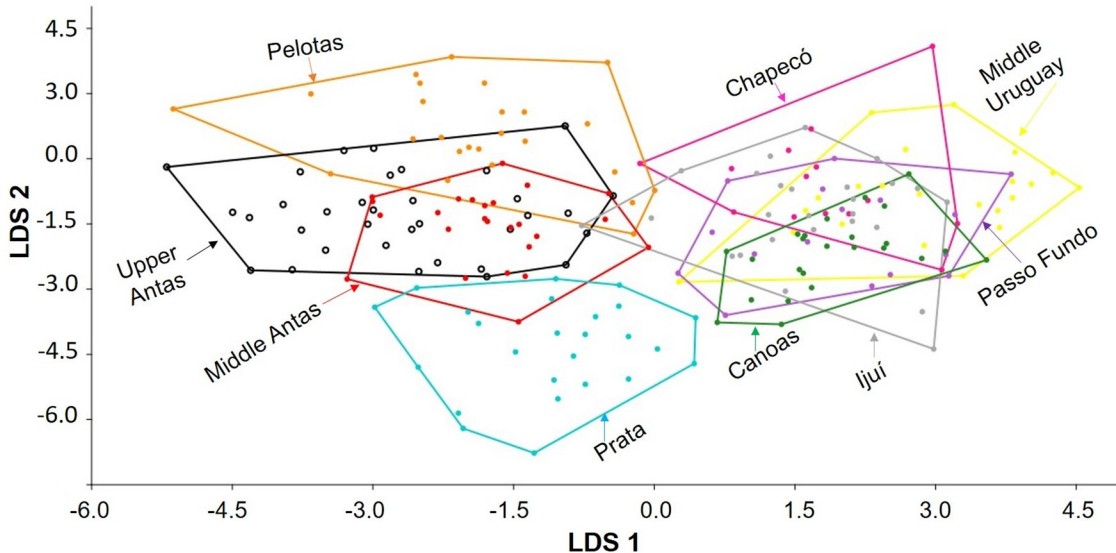

**Fig 10. Scatter plot of Linear discriminant analysis LDA1 x LDA2 factor scores.** Each color represents an area studied in the Taquari River and Uruguay River basins.

**Table 5. Haplotype and nucleotide data of *Pareiorhaphis hystrix*.coI.**

| Áreas | N | s | h | hd | π |
|---|---|---|---|---|---|
| T—Upper Antas | 7 | 3 | 2 | 0.28 +/- 0.19 | 0.04 +/- 0.04 |
| T—Middle Antas | 3 | 0 | 1 | 0.00 +/- 0.00 | 0.00 +/- 0.00 |
| T—Prata | 2 | 5 | 2 | 1.00 +/- 0.50 | 0.26 +/- 0.28 |
| U—Chapecó | 4 | 3 | 3 | 0.83 +/- 0.22 | 0.07 +/- 0.07 |
| U—Pelotas | 3 | 0 | 1 | 0.00 +/- 0.00 | 0.00 +/- 0.00 |
| U—Ijuí | 3 | 2 | 2 | 0.66 +/- 0.31 | 0.07 +/- 0.07 |
| U—Passo Fundo | 3 | 0 | 1 | 0.00 +/- 0.00 | 0.00 +/- 0.00 |
| U—Middle Uruguay | 3 | 3 | 2 | 0.66 +/- 0.31 | 0.10 +/- 0.09 |
| U—Canoas | 4 | 3 | 2 | 0.50 +/- 0.26 | 0.07 +/- 0.07 |
| *Cytb* | | | | | |
| T—Upper Antas | 7 | 7 | 4 | 0.80 +/- 0.12 | 0.04 +/- 0.03 |
| T—Middle Antas | 4 | 2 | 3 | 0.83 +/- 0.22 | 0.04 +/- 0.04 |
| T—Prata | 4 | 8 | 3 | 0.83 +/- 0.22 | 0.13 +/- 0.10 |
| U—Chapecó | 5 | 7 | 4 | 0.90 +/- 0.16 | 0.09 +/- 0.07 |
| U—Pelotas | 4 | 7 | 3 | 0.83 +/- 0.22 | 0.13 +/- 0.09 |
| U—Ijuí | 2 | 0 | 1 | 0.00 +/- 0.00 | 0.00 +/- 0.00 |
| U—Passo Fundo | 3 | 0 | 1 | 0.00 +/- 0.00 | 0.00 +/- 0.00 |
| U—Middle Uruguay | 4 | 9 | 3 | 0.83 +/- 0.22 | 0.16 +/- 0.12 |
| U—Canoas | 5 | 2 | 3 | 0.70 +/- 0.21 | 0.11 +/- 0.08 |
| **16S** | | | | | |
| T—Upper Antas | 7 | 0 | 1 | 0.00 +/- 0.00 | 0.00 +/- 0.00 |
| T—Middle Antas | 3 | 0 | 1 | 0.00 +/- 0.00 | 0.00 +/- 0.00 |
| T—Prata | 4 | 1 | 2 | 0.50 +/- 0.26 | 0.07 +/- 0.08 |
| U—Chapecó | 5 | 2 | 2 | 0.40 +/- 0.23 | 0.11 +/- 0.11 |
| U—Pelotas | 5 | 0 | 1 | 0.00 +/- 0.00 | 0.00 +/- 0.00 |
| U—Ijuí | 4 | 0 | 1 | 0.00 +/- 0.00 | 0.00 +/- 0.00 |
| U—Passo Fundo | 5 | 2 | 2 | 0.40 +/- 0.23 | 0.11 +/- 0.11 |
| U—Middle Uruguay | 5 | 1 | 2 | 0.60 +/- 0.17 | 0.08 +/- 0.09 |
| U—Canoas | 4 | 0 | 1 | 0.00 +/- 0.00 | 0.00 +/- 0.00 |
| **12S** | | | | | |
| T—Upper Antas | 5 | 4 | 3 | 0.80 +/- 0.16 | 0.14 +/- 0.11 |
| T—Middle Antas | 5 | 3 | 3 | 0.70 +/- 0.21 | 0.11 +/- 0.09 |
| T—Prata | 3 | 2 | 3 | 1.00 +/- 0.27 | 0.09 +/- 0.09 |
| U—Chapecó | 5 | 4 | 4 | 0.90 +/- 0.16 | 0.11+/- 0.09 |
| U—Pelotas | 5 | 1 | 2 | 0.40 +/- 0.23 | 0.02 +/- 0.03 |
| U—Ijuí | 2 | 0 | 1 | 0.00 +/- 0.00 | 0.00 +/- 0.00 |
| U—Passo Fundo | 3 | 0 | 1 | 0.00 +/- 0.00 | 0.00 +/- 0.00 |
| U—Middle Uruguay | 4 | 4 | 2 | 0.50 +/- 0.26 | 0.14 +/- 0.11 |
| U—Canoas | 4 | 1 | 2 | 0.50 +/- 0.26 | 0.03 +/- 0.04 |

Sample size (N); number of polymorphic sites (s), number of haplotypes (h), haplotypic diversity (hd), and nucleotide diversity (π), per individual gene *coI*, *cytb*, 16S and 12S. T = Taquari River basin, U = Uruguay River basin. Specimens with many missing data removed.

proximity with specimens from areas in the Taquari River basin, while Passo Fundo and Chapecó presented greater distance from the other areas. For each individual haplotype network (Fig 12), in general, haplotype sharing was observed between the different areas, although exclusive haplotypes are also observed. There is low differentiation between populations, with few mutations separating them.

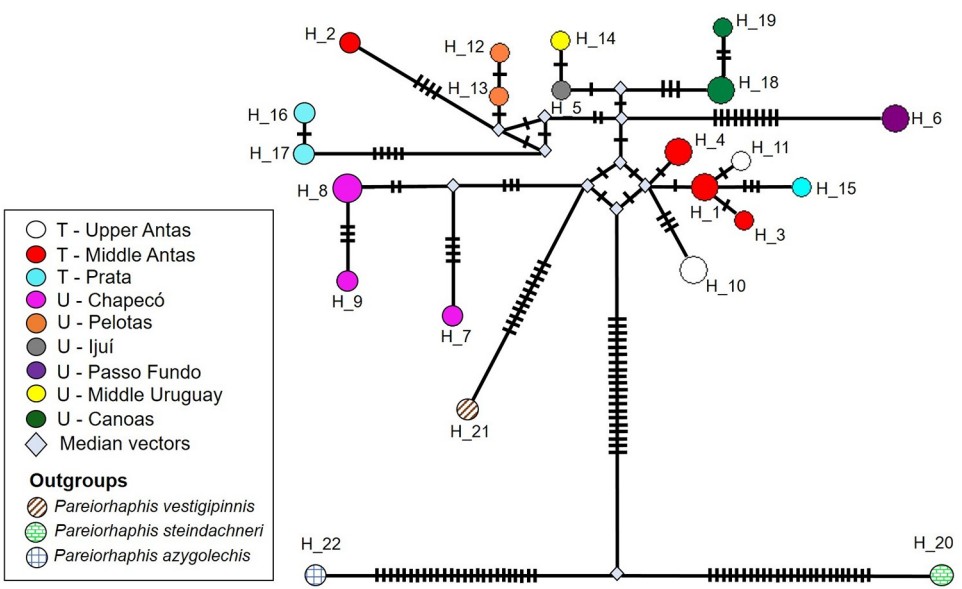

**Fig 11. Haplotype network of concatenated genes *coI*, *cytb*, 16S, and 12S for all samples of *Pareiorhaphis hystrix*.**
Transverse bars between haplotypes indicate number of mutations; size of each circle is proportional to the number of
specimens grouped in each haplotype. H = haplotype; T = Taquari River basin; U = Uruguay River basin; median
vectors = unsampled sequences. Specimens with large amount of missing data removed from analysis.

For *coI* (Fig 12) 12 haplotypes were observed in the network. There is a central haplotype
shared between the different river basins, from which several others depart, with 1–4 muta-
tional steps. The *cytb* (Fig 12) presented 23 haplotypes in the network, and a greater genetic
structure per area, with only haplotype 17 shared between the separate basins: Pelotas (Uru-
guay River basin) and Middle Antas (Taquari River basin). The 16S (Fig 12) showed little vari-
ation among specimens, consisting of only six haplotypes with few mutations between them
(1–3 mutational steps), with a central haplotype shared by all areas except Chapecó. Finally,
for the 12S (Fig 12), 15 haplotypes were obtained, the central haplotype being shared between
the two watersheds, originating several others with few mutations (1–5 steps).

## Genetic distance

The genetic distance of the concatenated data (Table 6) points to a species-level separation
with a percent above 3%, as observed for *Pareiorhaphis azygolechis* and *P. stendachneri*, or
approximately 1% for *P. vestigipinnis*. In general, each area of *P. hystrix*, when compared to the
others, presented genetic distance below 1%, with Passo Fundo (0.60 to 0.95%) and Chapecó
(0.66 to 0.95%) presenting highest genetic distance when compared to other areas. For *coI*
alone (Table 7), the genetic distance indicated a species-level separation ranging from 4–6%,
except for *P. vestigipinnis*, which also showed lower differentiation (less than 2%). The com-
parison among areas of *P. hystrix* never exceeded 1.37%, and Prata presented the greater
genetic distance relative to the other areas (some values higher than 1%).

## Phylogeny

The phylogenetic analysis of the concatenated genes revealed most basal branches with high
posterior probability (PP) values, and found *Pareiorhaphis hystrix* as monophyletic at 99% PP
(Fig 13). Chapecó grouped the specimens in a clade sister to all other areas (100% PP), while
Passo Fundo had a longer branch length when compared to the others, indicating a larger

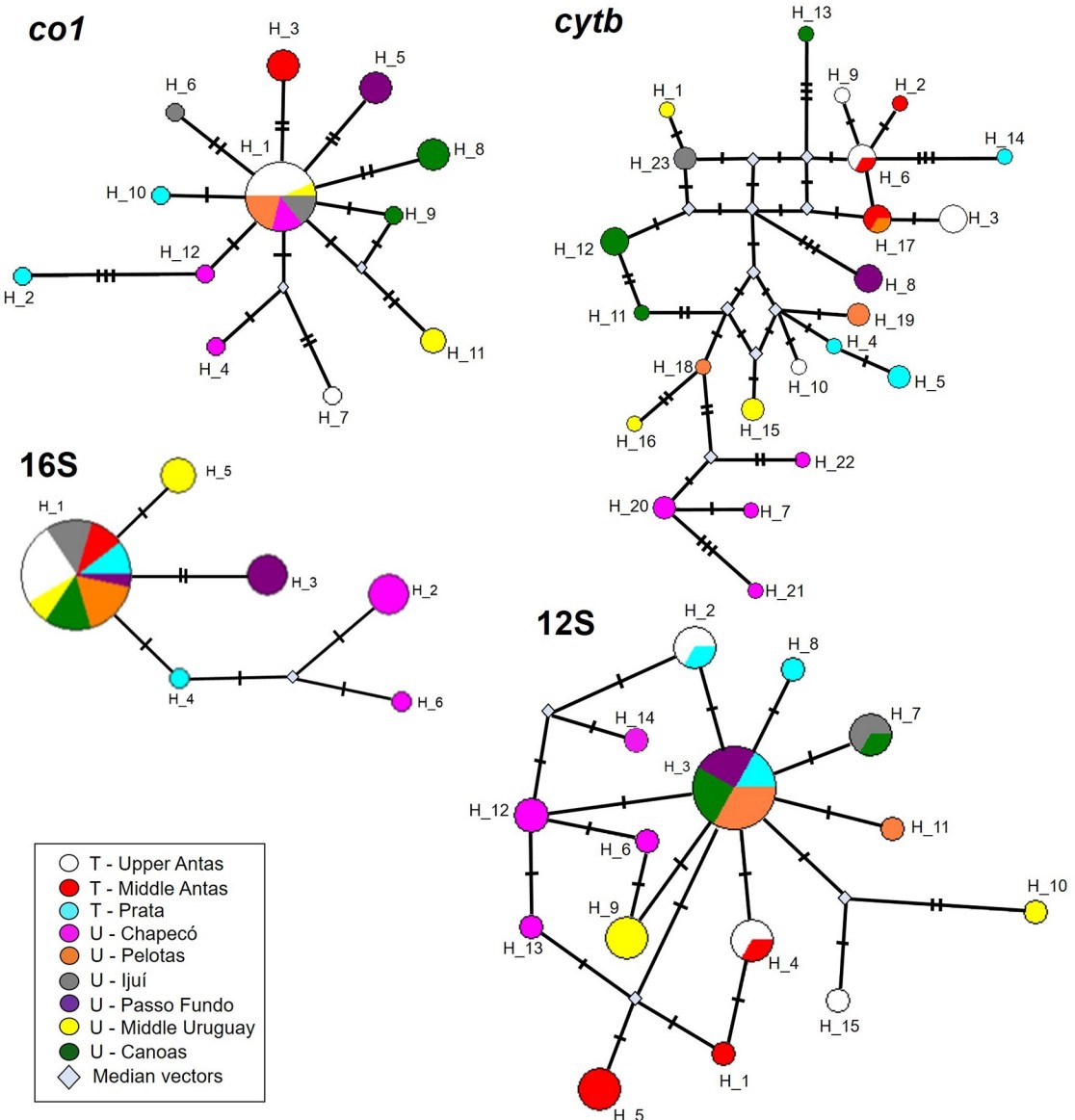

**Fig 12. Haplotype network of mitochondrial genes of *Pareiorhaphis hystrix*.** Transverse bars between haplotypes indicate number of mutations; size of each circle is proportional to the number of specimens grouped in each haplotype. H = haplotype; T = Taquari River basin; U = Uruguay River basin; median vectors = unsampled sequences. Specimens with large amount of missing data removed from analysis.

number of transformations. Despite some individual clades depict high support (96–100% PP), most internal nodes are weakly supported and highly polytomic, producing no separation between areas or even between watersheds.

## Coalescence analysis

The coalescence analysis—GMYC with the entire data matrix suggested the presence of 14 operational taxonomic units (OTUs), 12 of which grouping mixed populations from different areas and both river basins, one of which including specimens of *P. hystrix* and *P. vestigipinnis*. The maximum likelihood for the null model was 344.29, and 349.43 for the GMYC model,

**Table 6. Mean genetic distance between and within different areas of *Pareiorhaphis hystrix* and outgroup species for concatenated mitochondrial genes.**

| | | 1 | 2 | 3 | 4 | 5 | 6 | 7 | 8 | 9 | 10 | 11 | 12 | Within areas |
|---|---|---|---|---|---|---|---|---|---|---|---|---|---|---|
| 1 | T—Upper Antas | | 0.09 | 0.11 | 0.15 | 0.09 | 0.09 | 0.13 | 0.11 | 0.09 | 0.18 | 0.44 | 0.37 | 0.31 ± 0.08 |
| 2 | T—Middle Antas | 0.00 | | 0.12 | 0.16 | 0.10 | 0.10 | 0.14 | 0.13 | 0.10 | 0.20 | 0.46 | 0.38 | 0.21 ± 0.07 |
| 3 | T—Prata | 0.50 | 0.57 | | 0.17 | 0.09 | 0.11 | 0.14 | 0.11 | 0.10 | 0.18 | 0.43 | 0.38 | 0.43 ± 0.10 |
| 4 | U—Chapecó | 0.71 | 0.74 | 0.80 | | 0.14 | 0.16 | 0.17 | 0.16 | 0.15 | 0.19 | 0.43 | 0.38 | 0.36 ± 0.08 |
| 5 | U—Pelotas | 0.38 | 0.45 | 0.44 | 0.66 | | 0.10 | 0.12 | 0.09 | 0.08 | 0.18 | 0.44 | 0.38 | 0.32 ± 0.09 |
| 6 | U—Ijuí | 0.38 | 0.40 | 0.47 | 0.71 | 0.35 | | 0.13 | 0.10 | 0.08 | 0.19 | 0.42 | 0.38 | 0.13 ± 0.06 |
| 7 | U—Passo Fundo | 0.70 | 0.69 | 0.78 | 0.95 | 0.63 | 0.60 | | 0.12 | 0.12 | 0.20 | 0.45 | 0.38 | 0.42 ± 0.09 |
| 8 | U—Middle Uruguay | 0.52 | 0.63 | 0.60 | 0.81 | 0.42 | 0.44 | 0.75 | | 0.10 | 0.18 | 0.46 | 0.38 | 0.40 ± 0.09 |
| 9 | U—Canoas | 0.44 | 0.49 | 0.49 | 0.72 | 0.38 | 0.30 | 0.65 | 0.48 | | 0.18 | 0.44 | 0.37 | 0.22 ± 0.07 |
| 10 | *P. vestigipinnis* | 0.90 | 0.90 | 0.90 | 0.90 | 0.81 | 0.81 | 1.06 | 1.00 | 0.81 | | 0.46 | 0.40 | 0.06 ± 0.05 |
| 11 | *P. steindachneri* | 3.51 | 3.60 | 3.60 | 3.65 | 3.50 | 3.30 | 3.64 | 3.70 | 3.50 | 3.30 | | 0.48 | 0.00 ± 0.00 |
| 12 | *P. azygolechis* | 3.17 | 3.22 | 3.30 | 3.30 | 3.30 | 3.11 | 3.40 | 3.40 | 3.21 | 3.24 | 3.75 | | 0.00 ± 0.00 |

Values given as percent. All sequenced specimens were considered.

Standard deviation given above diagonal (between areas) and on right (within areas).

with significant difference (0.005*). When the genetically most distant outgroup, *P. steindachneri*, was removed (Fig 14), the GMYC model suggested three separate OTUs, with low support values. The first OTU was represented by specimens from different areas and basins, the second by specimens from Middle Uruguay, and the third by *P. vestigipinnis* and *P. hystrix* from Canoas and Middle Uruguay (Uruguay River basin), and Prata (Taquari River basin). The maximum likelihood for the null model was 337.37, and 345.52 for the GMYC model, with significant difference (0.0002**).

## Discussion

Specimens of *Pareiorhaphis hystrix* with both small and delicate or large and strong odontodes in the cheek fleshy lobe are common and were compared in this study. The cheek fleshy lobe also shows a large variation in size and shape. When the fleshy lobe is well developed, odontodes tend to be poorly visible, since they are structures attached to the bone and emerging

**Table 7. Mean genetic distance between and within different areas of *Pareiorhaphis hystrix* and outgroup species for *coI*.**

| | | 1 | 2 | 3 | 4 | 5 | 6 | 7 | 8 | 9 | 10 | 11 | 12 | Within areas |
|---|---|---|---|---|---|---|---|---|---|---|---|---|---|---|
| 1 | T—Upper Antas | | 0.32 | 0.43 | 0.19 | 0.14 | 0.23 | 0.32 | 0.29 | 0.43 | 0.60 | 1.53 | 1.24 | 0.33 ± 0.18 |
| 2 | T—Middle Antas | 0.66 | | 0.52 | 0.31 | 0.29 | 0.35 | 0.41 | 0.40 | 0.52 | 0.63 | 1.54 | 1.35 | 0.41 ± 0.24 |
| 3 | T—Prata | 1.09 | 1.13 | | 0.42 | 0.40 | 0.45 | 0.50 | 0.49 | 0.49 | 0.55 | 1.51 | 1.28 | 0.86 ± 0.36 |
| 4 | U—Chapecó | 0.37 | 0.64 | 0.97 | | 0.11 | 0.24 | 0.25 | 0.48 | 0.50 | 0.82 | 1.80 | 1.41 | 0.50 ± 0.28 |
| 5 | U—Pelotas | 0.20 | 0.47 | 0.89 | 0.16 | | 0.18 | 0.28 | 0.26 | 0.41 | 0.58 | 1.52 | 1.28 | 0.16 ± 0.16 |
| 6 | U—Ijuí | 0.46 | 0.72 | 1.15 | 0.45 | 0.26 | | 0.34 | 0.31 | 0.45 | 0.62 | 1.55 | 1.30 | 0.30 ± 0.21 |
| 7 | U—Passo Fundo | 0.49 | 0.78 | 1.18 | 0.39 | 0.29 | 0.55 | | 0.39 | 0.50 | 0.66 | 1.57 | 1.31 | 0.68 ± 0.27 |
| 8 | U—Middle Uruguay | 0.69 | 0.91 | 1.37 | 0.97 | 0.52 | 0.75 | 0.83 | | 0.48 | 0.53 | 1.50 | 1.33 | 0.48 ± 0.25 |
| 9 | U—Canoas | 0.88 | 1.17 | 1.28 | 0.95 | 0.68 | 0.95 | 0.98 | 1.05 | | 0.52 | 1.51 | 1.24 | 0.48 ± 0.27 |
| 10 | *P. vestigipinnis* | 1.03 | 1.28 | 1.19 | 1.28 | 0.82 | 1.10 | 1.13 | 1.09 | 0.92 | | 1.38 | 1.29 | 0.00 ± 0.00 |
| 11 | *P. steindachneri* | 5.91 | 6.11 | 6.02 | 6.43 | 5.69 | 5.98 | 6.01 | 5.88 | 5.79 | 4.77 | | 1.40 | 0.00 ± 0.00 |
| 12 | *P. azygolechis* | 4.01 | 4.62 | 4.53 | 4.28 | 4.01 | 4.29 | 4.32 | 4.50 | 4.10 | 4.05 | 4.90 | | 0.00 ± 0.00 |

Standard deviation given above diagonal (between areas) and on right (within areas). Values given as percent. All sequenced specimens considered.

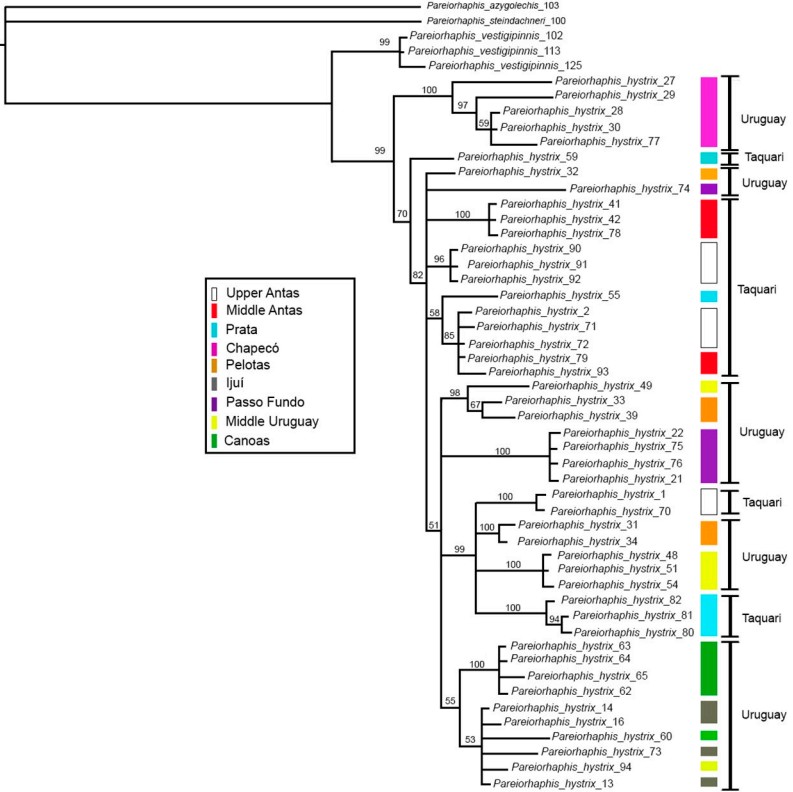

**Fig 13. Bayesian phylogenetic reconstruction based on concatenated mitochondrial genes of *Pareiorhaphis hystrix* (*coI*, *cytb*, 16S, 12S).** Color bars represent the nine geographic areas studied; black bars indicate river basin. Node numbers represent posterior probability.

through the fleshy lobe. Both the variation in odontodes and the associated fleshy lobe did not show any association with area or river drainage. Similarly, observed variation in coloration does not seem to be related with area or river basin. The color pattern of the species as described by Pereira & Reis [33], with the dorsal surface of body and head dark gray, sometimes brownish gray, covered by dark brown to black blotches, and the ventral region characterized by a light, pale yellow, presented variation between the specimens studied. This variation is expected at intraspecific level and is not sufficient to distinguish specimens at interspecific level.

The original description of the species [33] did not mention the presence of plates in the anterior region of the abdomen as a characteristic of *Pareiorhaphis hystrix*, and the presence of such plates is a novelty herein described. Some species such as *P. nasuta* Pereira, Vieira & Reis, 2007 and *P. scutula* Pereira, Vieira & Reis, 2010, described from the Doce River basin, *P. ruschii* Pereira, Lehmann & Reis, 2012, from tributaries of the Piraquê-Açu and Reis Magos rivers, and *P. parmula* Pereira, 2005, from the Iguaçu River basin, possess minute plates embedded in the abdominal skin. These species have minute plates distributed on each side of the pectoral girdle, right posterior to the gill opening, and *P. scutula* also has plates scattered throughout the abdominal region, from the pectoral girdle to the insertion of the pelvic fin. The function of such plates is still unknown, but specimens possessing a high concentration of plates in the abdomen may have an advantage when fixing themselves to stones in a strong water current, a common environment for these loricariids. In *P. hystrix* abdominal plates occur in individuals from the Uruguay River basin except the population from Pelotas, which

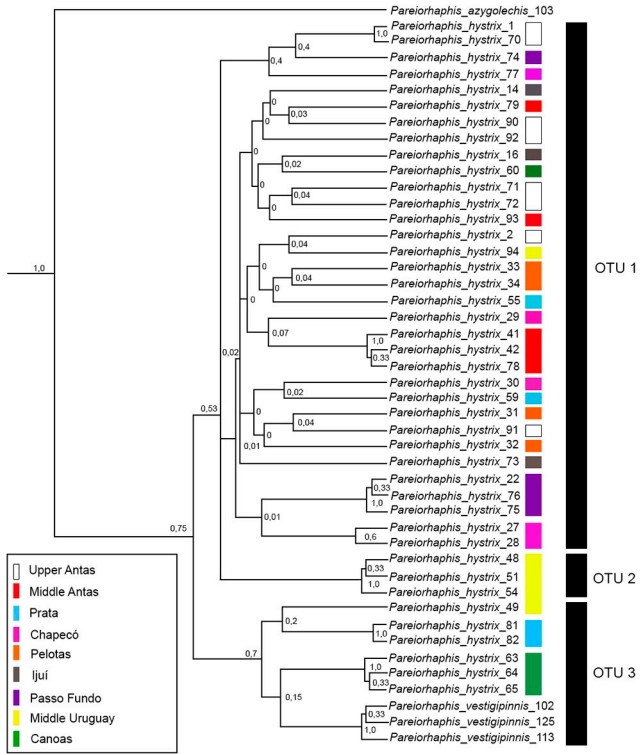

**Fig 14. Bayesian topology and species delimitation estimated using the single-threshold in the General Mixed Yule Coalescent model (GMYC) for *Pareiorhaphis hystrix* (*coI*).** Color bars represent the nine geographic areas studied; black bars represent possible OTUS. Node numbers correspond to BI posterior probability (PP). OTU = operational taxonomic unit.

is devoid of plates like the populations from the Taquari River. This pattern is corroborated by some results of the ANOVA and by the LDA which also indicated a closer similarity of the Pelotas and the areas in the Taquari River basin.

The significant variation ANOVA found for some counts, such as the number of premaxillary and dentary teeth, may be due to the fact that specimens often lose and replace their teeth and may be in different stages of tooth replacement [34]. Although significant values were observed, the differences between the counts were not discrete, fitting within the expected variation for a species.

The star-shaped individual haplotype networks with a central haplotype and additional haplotype starting with a few mutations (genes *coI*, 16S, 12S), may suggest a recent expansion of the parental haplotype [35–38]. Moreover, a high number of haplotypes was found considering the relatively small sample in some of the areas, followed by the small nucleotide diversity–few mutations among haplotypes, also corroborating the idea of a recent expansion [39, 40]. The *cytb* gene is the most differentiated between the areas, with each area carrying its own haplotypes. The concatenated network, on the other hand, lost the star pattern observed individually, with the areas having their own haplotypes, although the distance between them still being small. In general, the most different areas are Chapecó and Passo Fundo, in agreement with the genetic distance and the concatenated phylogeny obtained.

Recently, Lima et al. [41] conducted a population analysis of *Pareiorhaphis garbei* in the Macaé and Macacu rivers, southeastern Brazil, where they presented haplotype networks generated for *coI* and *cytb*. No haplotype was shared among populations with a large number of

mutational steps between populations. Contrarily, in the present study the haplotypes of the separated populations are very close, and there is still a large internal variability within the areas. Although there is not yet enough morphological evidence indicating a different species restricted to the upper-middle portion of the Uruguay basin, a slight differentiation by geographic area is already observed, especially in the Chapecó and Passo Fundo areas.

Although the coalescence species delimitation analysis suggests the existence of separate OTUs, this separation varies largely with the removal of the most genetically distant outgroup, suggesting the revealed genetic structure represents an artifact, a conclusion also supported by the groups not being observed iteratively in this study. The genetic distance for *coI* fails to indicate the presence of multiple species, considering the 2% threshold suggested to identify separate species lineages [42–45]. Higher values of genetic differentiation observed for Prata and Canoas specimens may have influenced the identification of OTUs in the coalescence analysis.

A member of the outgroup, *Pareiorhaphis vestigipinnis*, is poorly differentiated from *P. hystrix*. The main distinguishing morphological features of the former species are the absence of an adipose fin, which is always present and well-developed in *P. hystrix*, and the club-shaped pectoral-fin spine [46], which is homogeneously wide in the latter. While the phylogeny recovered *P. vestigipinnis* as sister to *P. hystrix*, corroborating the results of Pereira & Reis [8], it was found more closely related, despite weakly supported, to *P. hystrix* from Canoas, Prata, and Middle Uruguay in the coalescence analysis. Moreover, the concatenated haplotype network corroborates the phylogeny, where a greater separation is observed between *P. vestigipinnis* and all populations of *P. hystrix*. The 2% interspecific differentiation threshold commonly used for *coI* does not appear to apply to *P. vestigipinnis* and *P. hystrix*, which have genetic divergence with values even below 1% when compared to some *P. hystrix* populations, while *P. azygolechis* and *P. steindachneri* show distances above 3%. Despite the low genetic distance, two morphological features consistently distinguish *P. vestigipinnis* from *P. hystrix*, and they are thus maintained as a separate species.

While morphology indicates a slight trend of watershed separation, this was not corroborated by the molecular data. In general, haplotype sharing occurs between the two river basins with few mutations separating the haplotypes. However, some specimens from Pelotas presented greater proximity to those from Taquari, similarly to the morphology. A closer proximity of the individuals from Pelotas, which belongs to the Uruguay River basin, to specimens from the adjacent Taquari River basin, could be possibly explained by headwater capture events [47–51]. For example, the trichomycterid catfish *Cambeva balios* (Ferrer & Malabarba, 2013) occurs in the upper portions of the tributaries of the Antas and Caí rivers of the Taquari basin, and in the headwaters of the coastal Mampituba River. Such occurrence at the headwaters of two unconnected drainages could be possibly explained by headwater capture events [48]. The same phenomenon is reported for *Astyanax brachypterygium* Bertaco & Malabarba, 2001, which occurs in the upper portion of the Uruguay, Jacui, and Taquari basins, and *Cambeva poikilos* (Ferrer & Malabarba, 2013), with specimens at the upper portions of the Taquari and Jacuí tributaries. Headwater capture is known for moving a creek from one river basin to another, and thus mixing different species groups, considerably increasing regional levels of interspecific richness [52].

Although *Pareiorhaphis hystrix* is widely distributed in the Uruguay and upper Taquari river basins, this pattern does not seem to be common to other loricariids. The opposite situation occurs in other loricariid genus sharing most of the same geographic distribution of *P. hystrix*, the hypoptopomatine *Eurycheilichthys*. The type species, *E. pantherinus* (Reis & Schaefer, 1992) is widely distributed in the upper reaches of the Uruguay River, *E. limulus* Reis & Schaefer, 1998 is endemic to the upper portions of the Jacuí River, and seven species occur in

tributaries of the Taquari River [53]. Contrary to *Eurycheilichthys*, the present analyses indicate that *P. hystrix* retains old haplotypes common to both watersheds, showing little differentiation between the specimens in these geographical areas. If these individuals were separated long enough, there would probably become different species distributed along the two watersheds. Natural selection seems to be acting differently on the morphology and the genetics of *P. hystrix*, as the iterative analyses suggest a single, although phenotypically variable species, thus not refuting the null hypothesis.

## Supporting information

**S1 Table. Information of specimens used in morphological and molecular analyses of *Pareiorhaphis*.** Specimens used for morphological analyses are marked MORPH in the Gene column.
(DOCX)

## Acknowledgments

We are deeply indebted to Carlos Lucena and Margarete Lucena for continued assistance and support provided at the MCP Laboratory of Ichthyology. We thank Juliana Wingert and Luiz Malabarba (UFRGS), and Pablo Lehmann (UNISINOS), for loaning fishes and donating tissue samples under their care for DNA extraction. This study was part of the master's thesis of the first author (PCF), who is grateful to Maria Laura Delapieve, Vanessa Meza, Dario Faustino and, especially, Bárbara Calegari, for the invaluable help in learning methods and conducting molecular analyzes. Maria Laura Delapieve also critically reviewed the manuscript. PCF thanks the laboratory colleagues Alejandro Londoño, Álvaro Neto, Arthur Capelli, Rafael Lugo and Suelen Gamarra for the conversations, camaraderie, and ideas exchanged, as well as Cláudio Zawadzki, Lenice Souza-Shibatta, and Ricardo Benine, members of the evaluation committee of the thesis for suggestions.

## Author Contributions

**Conceptualization:** Patrícia C. Fagundes, Roberto E. Reis.

**Data curation:** Patrícia C. Fagundes.

**Formal analysis:** Patrícia C. Fagundes.

**Funding acquisition:** Roberto E. Reis.

**Investigation:** Patrícia C. Fagundes, Roberto E. Reis.

**Methodology:** Edson H. L. Pereira, Roberto E. Reis.

**Project administration:** Roberto E. Reis.

**Resources:** Roberto E. Reis.

**Supervision:** Edson H. L. Pereira, Roberto E. Reis.

**Validation:** Patrícia C. Fagundes, Edson H. L. Pereira, Roberto E. Reis.

**Writing – original draft:** Patrícia C. Fagundes.

**Writing – review & editing:** Edson H. L. Pereira, Roberto E. Reis.

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
