## [Decision Letter · Decision Letter 0]

26 Jun 2020

PONE-D-20-14256

Iterative taxonomic study of Pareiorhaphis hystrix (Siluriformes, Loricariidae) suggests a single, yet phenotypically variable, species in south Brazil

PLOS ONE

Dear Dr. Reis,

Thank you for submitting your manuscript to PLOS ONE. After careful consideration, we feel that it has merit but does not fully meet PLOS ONE’s publication criteria as it currently stands. Therefore, we invite you to submit a revised version of the manuscript that addresses the points raised during the review process.

We look forward to receiving your revised manuscript.

Kind regards,

Feng ZHANG, Ph.D.

Academic Editor

PLOS ONE

Journal Requirements:

Additional Editor Comments (if provided):

Reviewers' comments:

Reviewer's Responses to Questions

**Comments to the Author**

1. Is the manuscript technically sound, and do the data support the conclusions?

Reviewer #1: Yes

Reviewer #2: Yes

2. Has the statistical analysis been performed appropriately and rigorously? 

Reviewer #1: Yes

Reviewer #2: Yes

3. Have the authors made all data underlying the findings in their manuscript fully available?

Reviewer #1: Yes

Reviewer #2: Yes

4. Is the manuscript presented in an intelligible fashion and written in standard English?

Reviewer #1: Yes

Reviewer #2: Yes

5. Review Comments to the Author

Reviewer #1: This is an important contribution to the understanding of Neotropical Fishes. The study has up-to-date methods that were implemented comprehensively to test the hypotheses proposed in the beginning of the study. Integrative taxonomic studies, as this one, that integrate morphology and genetic data from multiple genes are the best way to test species con-specificity. The results are also very detailed described and illustrated and support the conclusion that Pareiorhaphis hystrix is composed on a single, although variable, species. My main critic suggestion is that the results of the study, especially concerning the haplotype analysis, the OUT analysis, and the coI genetic distance, support that Pareiorhaphis vestigipinnis is likely co-specific with P. hystrix. In addition, the geographic distribution of the two species are complementary, and the reduction of adipose fin is a character that is variable in other armored catfishes, and that may also be variable in this case. Authors should possibly change the paper to include the synonymization. If not, they should argue more what they defer not doing that. My only other suggestions: 1. Merge figures 1 and 2 into a single plate. 2. Why Passo Fundo population does not have a H number in Fig. 11?

Reviewer #2: The manuscript provides an important contribution to the taxonomy and evolutionary relationships of Pareiorhaphis populations from Uruguay and Jacui basins. The authors used multiple lines of evidence to investigate the identity and distribution of Pareiorhaphis hystrix. In general , they have enough result supporting their hypothesis but some issues need to be fixed before the publication. See the following items.

Major issues:

1. The authors should revise the language to improve readability.

2. The authors should rewrite the introduction to ensure that the readers understand the importance of the study and the reasons why they chose the concept of iterative taxonomy instead of alternative approaches traditionally used in fish taxonomy.

3. In Material and Methods, please, include PCR protocols of each gene as well as the sequencing procedures.

4. In the Results (line 194) it is mentioned that more than one pattern of development of cheek odontodes and associated lateral fleshy lob was observed in males. Figure 4 illustrates eleven patterns but there is no explanation of how to distinguish one from other.

5. The name "abdominal plates" (Line 225) sounds inappropriate for the region where these platelets occur. If the plates occur only on the pectoral girdle, consider changing the name to "pectoral plates" or "thoracic plates" updating the text where necessary.

6. PCA and LDA showed quite similar results and they are not crucial for taxonomic decisions. They just show in different ways that there are no significant differences in morphology between populations. I recommend excluding one of these analyses to avoid redundancies. Otherwise, explain clearly in Material and Methods the differences between the analyses and why the inclusion of both methods are important for the work.

7. Review carefully the Haplotype network results. Contrary to what is explained in the text (lines 344-347), I did not see any haplotype been shared between regions in Figure 11 (maybe the phrase is in the wrong place in the text). In addition, in line 366 is commented that only haplotype 17 is shared but in Figure 12 there is another haplotype being shared (H_6).

8. The authors have several interesting results but they are poorly explored and discussed. For instance, Table 5 and Figures 11 and 12 show that Ijui and Passo Fundo are the regions with less number of haplotypes. Besides, they commented that the COI 2% threshold did not work to distinguish species, but there are other works showing that in some cases the threshold value might be larger or lower (e. g. Hypostomus) than 2%.

Minor issues:

1. In line 64, the authors comment that Pareiorhaphis is monophyletic citing a morphology-based study. Since the authors use molecular data in their phylogenetic analysis, it is worth also informing whether molecular phylogenies for the genus are available and, in positive cases, if these works corroborate with morphology-based studies.

2. Inform the distribution of the genus in the Introduction and the habitats explored by its species.

3. Line 67 has a missing reference. Please, check carefully all cited references.

4. Line 429: "time-divergence coalescence"? Looks like it is missing some information in the Figure.

6. PLOS authors have the option to publish the peer review history of their article (what does this mean?). If published, this will include your full peer review and any attached files.

Reviewer #1: **Yes: **Jose Birindelli

Reviewer #2: No

---

## [Author Response · Author response to Decision Letter 0]

7 Jul 2020

Detailed response to all comments by editor and reviewers are uploaded in the file "Response to reviewers"

---

## [Editor Report · Decision Letter 1]

22 Jul 2020

Iterative taxonomic study of Pareiorhaphis hystrix (Siluriformes, Loricariidae) suggests a single, yet phenotypically variable, species in south Brazil

PONE-D-20-14256R1

Dear Dr. Reis,

We’re pleased to inform you that your manuscript has been judged scientifically suitable for publication and will be formally accepted for publication once it meets all outstanding technical requirements.

Kind regards,

Feng ZHANG, Ph.D.

Academic Editor

PLOS ONE
---

## [Editor Report · Acceptance letter]

20 Aug 2020

PONE-D-20-14256R1 

Iterative taxonomic study of *Pareiorhaphis hystrix* (Siluriformes, Loricariidae) suggests a single, yet phenotypically variable, species in south Brazil. 

Dear Dr. Reis:

I'm pleased to inform you that your manuscript has been deemed suitable for publication in PLOS ONE. Congratulations! Your manuscript is now with our production department. 

Kind regards, 

on behalf of

Dr. Feng ZHANG 

Academic Editor

PLOS ONE